# DATA AUGMENTATION ALONE CAN IMPROVE ADVERSARIAL TRAINING

**Lin Li & Michael Spratling**
Department of Informatics
King's College London
30 Aldwych, London, WC2B 4BG, UK
{lin.3.li, michael.spratling}@kcl.ac.uk

## ABSTRACT

Adversarial training suffers from the issue of robust overfitting, which seriously impairs its generalization performance. Data augmentation, which is effective at preventing overfitting in standard training, has been observed by many previous works to be ineffective in mitigating overfitting in adversarial training. This work proves that, contrary to previous findings, data augmentation alone can significantly boost accuracy and robustness in adversarial training. We find that the hardness and the diversity of data augmentation are important factors in combating robust overfitting. In general, diversity can improve both accuracy and robustness, while hardness can boost robustness at the cost of accuracy within a certain limit and degrade them both over that limit. To mitigate robust overfitting, we first propose a new crop transformation, Cropshift, which has improved diversity compared to the conventional one (Padcrop). We then propose a new data augmentation scheme, based on Cropshift, with much improved diversity and well-balanced hardness. Empirically, our augmentation method achieves the state-of-the-art accuracy and robustness for data augmentations in adversarial training. Furthermore, when combined with weight averaging it matches, or even exceeds, the performance of the best contemporary regularization methods for alleviating robust overfitting. Code is available at: https://github.com/TreeLLi/DA-Alone-Improves-AT.

## 1 INTRODUCTION

Adversarial training, despite its effectiveness in defending against adversarial attack, is prone to overfitting. Specifically, while performance on classifying training adversarial examples improves during the later stages of training, test adversarial robustness degenerates. This phenomenon is called *robust overfitting* (Rice et al., 2020). To alleviate overfitting, Rice et al. (2020) propose to track the model's robustness on a reserved validation data and select the checkpoint with the best validation robustness instead of the one at the end of training. This simple technique, named early-stopping (ES), matches the performance of contemporary state-of-the-art methods, suggesting that overfitting in adversarial training impairs its performance significantly. Preventing robust overfitting is, therefore, important for improving adversarial training.

Data augmentation is an effective technique to alleviate overfitting in standard training, but it seems to not work well in adversarial training. Almost all previous attempts (Rice et al., 2020; Wu et al., 2020; Gowal et al., 2021a; Rebuffi et al., 2021; Carmon et al., 2019) to prevent robust overfitting by data augmentation have failed. Specifically, this previous work found that several advanced data augmentation methods like Cutout (DeVries & Taylor, 2017), Mixup (Zhang et al., 2018) and Cutmix (Yun et al., 2019) failed to improve the robustness of adversarially-trained models to match that of the simple augmentation Flip-Padcrop with ES, as shown in Fig. 1. Thus the method of using ES with Flip-Padcrop has been widely accepted as the "baseline" for combating robust overfitting. Even with ES, Cutout still fails to improve the robustness over the baseline, while Mixup boosts the robustness marginally ($< 0.4\%$) (Rice et al., 2020; Wu et al., 2020). This contrasts with their excellent performance in standard training. Recently, Tack et al. (2022) observed that AutoAugment (Cubuk et al., 2019) can eliminate robust overfitting and boost robustness greatly. This, however,

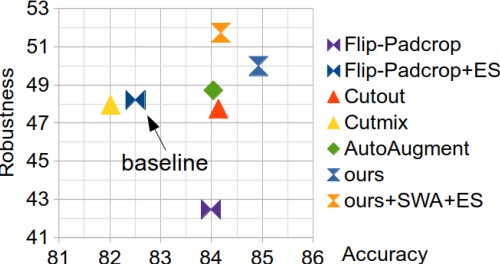

Figure 1: Our method is the only one that significantly improves both accuracy and robustness over the baseline (Flip-Padcrop with early-stopping). Cutout and Cutmix fail to beat the baseline regarding robustness. AutoAugment achieves only a small improvement on robustness over the baseline. Robustness is evaluated against AutoAttack. See Section 5 for details of training and evaluation settings.

contradicts the result of Gowal et al. (2021a); Carmon et al. (2019) where the baseline was found to outperform AutoAugment in terms of robustness. Overall, to date, there has been no uncontroversial evidence showing that robust generalization can be further improved over the baseline by data augmentation alone, and no convincing explanation about this ineffectiveness.

This work focuses on improving the robust generalization ability of adversarial training by data augmentation. We first demonstrate that the superior robustness of AutoAugment claimed by Tack et al. (2022) is actually a false security since its robustness against the more reliable AutoAttack (AA) (Croce & Hein, 2020) (48.71%) is just slightly higher than the baseline's (48.21%) as shown in Fig. 1 (see Appendix A for a detailed discussion). We then investigate the impact of the hardness and diversity of data augmentation on the performance of adversarial training. It is found that, in general, hard augmentation can alleviate robust overfitting and improve the robustness but at the expense of clean accuracy within a certain limit of hardness. Over that limit, both robustness and accuracy decline, even though robust overfitting is mitigated more with the increase in hardness. On the other hand, diverse augmentation generally can alleviate robust overfitting and boost both accuracy and robustness. These results give us the insight that the optimal data augmentation for adversarial training should have as much diversity as possible and well-balanced hardness.

To improve robust generalization, we propose a new image transformation, Cropshift, a more diverse replacement for the conventional crop operation, Padcrop. Cropshift is used as a component in a new data augmentation scheme that we call Improved Diversity and Balanced Hardness (IDBH). Empirically, IDBH achieves the state-of-the-art robustness and accuracy among data augmentation methods in adversarial training. It improves the end robustness to be significantly higher than the robustness of the baseline augmentation with early-stopping (Fig. 1), which all previous attempts failed to achieve. Furthermore, it matches the performance of the state-of-the-art regularization methods for improving adversarial training and, when combined with weight averaging, considerably outperforms almost all of them in terms of robustness.

## 2 RELATED WORKS

Robust overfitting can be successfully mitigated by smoothing labels, using Knowledge Distillation (KD) (Chen et al., 2021) and Temporal Ensembling (TE) (Dong et al., 2022), and/or smoothing weights using Stochastic Weight Averaging (SWA) (Chen et al., 2021) and Adversarial Weight Perturbation (AWP) (Wu et al., 2020). Moreover, Singla et al. (2021) found that using activation functions with low curvature improved the generalization of both accuracy and robustness. Alternatively, Yu et al. (2022) attributed robust overfitting to the training examples with small loss value, and showed that enlarging the loss of those examples during training, called Minimum Loss Constrained Adversarial Training (MLCAT), can alleviate robust overfitting. Our work prevents robust overfitting by data augmentation, and hence complements the above methods.

To date, it is still unclear if more training data benefits generalization in adversarial training. Schmidt et al. (2018) showed that adversarial training requires more data, compared to its standard training counterpart, to achieve the same level of generalization. In contrast, Min et al. (2021); Chen et al. (2020) proved that more training data can hurt the generalization in some particular adversarial training regimes on some simplified models and tasks. Empirically, a considerable improvement has been observed in both clean and robust accuracy when the training set is dramatically expanded, in a semi-supervised way, with unlabeled data (Carmon et al., 2019; Alayrac et al., 2019), e.g., using Robust Self-Training (RST) (Carmon et al., 2019) or with synthetic data generated by a generative

model (Gowal et al., 2021b). Although data augmentation alone doesn't work well, it was observed to improve robustness to a large degree when combined with SWA (Rebuffi et al., 2021) or Consistency (CONS) regularization (Tack et al., 2022). In contrast, our work doesn't require any additional data or regularization: it improves robust generalization by data augmentation alone.

Common augmentations (He et al., 2016a) used in image classification tasks include Padcrop (padding the image at each edge and then cropping back to the original size) and Horizontal Flip. Many more complicated augmentations have been proposed to further boost generalization. Cutout (DeVries & Taylor, 2017) and Random Erasing (Zhong et al., 2020) randomly drop a region in the input space. Mixup (Zhang et al., 2018) and Cutmix (Yun et al., 2019) randomly interpolate two images, as well as their labels, into a new one. AutoAugment (Cubuk et al., 2019) employs a combination of multiple basic image transformations like Color and Rotation and automatically searches for the optimal composition of them. TrivialAugment (Müller & Hutter, 2021) matches the performance of AutoAugment with a similar schedule yet without any explicit search, suggesting that this computationally expensive process may be unnecessary. The method proposed here improves on the above methods by specifically considering the diversity and hardness of the augmentations. The difference between data augmentation in standard and adversarial training is discussed in Appendix B.

## 3   How Data Augmentation Alleviates Robust Overfitting

This section describes an investigation into how the hardness and the diversity of data augmentation effects overfitting in adversarial training. During training, the model's robustness was tracked at each epoch using PGD10 applied to the test set. The checkpoint with the highest robustness was selected as the "best" checkpoint. Best (end) robustness/accuracy refers to the robustness/accuracy of the best (last) checkpoint. In this section, the terms accuracy and robustness refer to the end accuracy and robustness unless specified otherwise. The severity of robust overfitting was measured using the best robustness minus the end robustness. Hence, the more positive this gap in robustness the more severe the robust overfitting. The training setup is described in Appendix C.

### 3.1   Hardness

Hardness was measured by the Affinity metric (Gontijo-Lopes et al., 2021) adapted from standard training:

$$hardness = \frac{Robustness(M, D_{test})}{Robustness(M, D'_{test})} \tag{1}$$

where $M$ is an arbitrary model adversarially trained on the unaugmented training data. $D_{test}$ refers to the original test data set and $D'_{test}$ is $D_{test}$ with the augmentation (to be evaluated) applied. $Robustness(M, D)$ is the robust accuracy of $M$ evaluated using PGD50 on $D$. Hardness is a model-specific measure. It increases as the augmentation causes the data to become easier to attack, i.e., as the perturbed, augmented, data becomes more difficult to be correctly classified.

We found that moderate levels hardness can alleviate robust overfitting and improve the robustness but at the price of accuracy. Further increasing hardness causes both accuracy and robustness to decline, even though robust overfitting is alleviated further. The value of hardness where this occurs is very sensitive to the capacity of the model. Therefore, to maximize robustness, hardness should be carefully balanced, for each model, between alleviating robust overfitting and impairing accuracy.

**Experimental design.** We investigated the effects of hardness in adversarial training for individual and composed augmentations. For the individual augmentations the following 12 image transformations were choosen: ShearX, ShearY, TranslateX, TranslateY, Rotate, Color, Sharpness, Brightness, Contrast, Solarize, Cutout and Cropshift (a variant of Padcrop introduced in Section 4). For each Eq. (1) was used to calibrate the strength of the augmentation (e.g. angle for Rotation) onto one of 7 levels of hardness (see Appendix C.2 for specific values), except for Color and Sharpness which were applied at 3 strengths. For simplicity, the integers 1 to 7 are used to represent these 7 degrees of hardness. Standard Cutout is allowed to cut partially outside the image, and thus the hardness is not always directly related to the nominal size of the cut-out region (strength). To ensure alignment between strength and hardness, we force Cutout to cut only inside the image and refer this variant as Cutout-i. To control for diversity, each augmentation was applied with one degree of diversity. As a

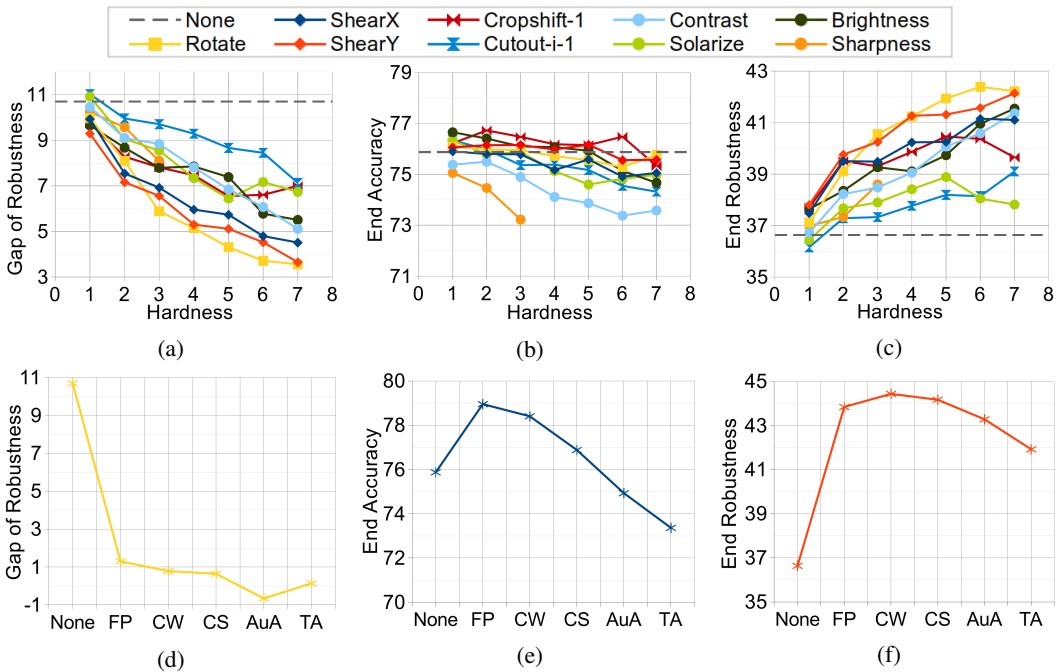

Figure 2: The performance of models trained with different individual transformations (top row) and composed augmentations (bottom row). None refers to no data augmentation applied. Robustness is evaluated against PGD50.

result the effects of applying Cutout and Cropshift with a certain hardness is deterministic throughout training. Specifically, Cutout always cuts at the fixed location (sampled once at the beginning of training). Similarly, CropShift crops a fixed region and shifts it to the fixed location (both sampled once at the beginning of training). We name these two variants as Cutout-i-1 and CropShift-1 respectively. Models were trained with each transformation at each hardness.

To investigate composed augmentations, models were trained with various multi-layered data augmentations: Flip-Padcrop (FP), FP-Cutout[Weak] (CW), FP-Cutout[Strong] (CS), FP-AutoAugment-Cutout (AuA) and FP-TrivialAugment-Cutout (TA). All of them shared the same parameters for Flip and Padcrop. CW and CS used 8x8 and 10x10 Cutout respectively. AuA and TA used 16x16 Cutout as in their original settings (Cubuk et al., 2019; Müller & Hutter, 2021). Different from the default experimental setting, augmentations here were always applied during training, and robustness was evaluated against AA since AuA was observed to fool the PGD attack (Appendix A).

Hardness increases from FP to TA as more layers stack up and/or the strength of individual components increases. Hence, this experiment can, as for the experiments with individual augmentations, be seen as an investigation into the effects of increasing hardness. Here, we are not controlling for diversity, which also roughly increases from FP to TA. However, this does not affect our conclusions as diversity boosts accuracy and robustness (see Section 3.2), and hence, the decrease in these values that we observe with increased hardness cannot be explained by the effects of increased diversity.

**Observations.** It can be seen that the gap between best and end robustness drops, i.e., robust overfitting turns milder with the increase in hardness in Figs. 2a and 2d. The gap of robustness for AuA is negative in Fig. 2d because the PGD10 attack was fooled to select a vulnerable checkpoint as the best: see Appendix A for more discussion. For accuracy and robustness, there are roughly three phases. First, both accuracy (Fig. 2b) and robustness (Fig. 2c) increase with hardness. This is only observed for some transformations like Cropshift at hardness 1 and 2. In this stage, the underlying model has sufficient capacity to fit the augmented data so it benefits from the growth of data complexity.

Second, accuracy (Fig. 2b) starts to drop while robustness (Fig. 2c) continuously increases. As the intensity of the transformation increases, the distribution of the transformed data generally deviates more from the distribution of the original data causing the mixture of them to be harder to fit for

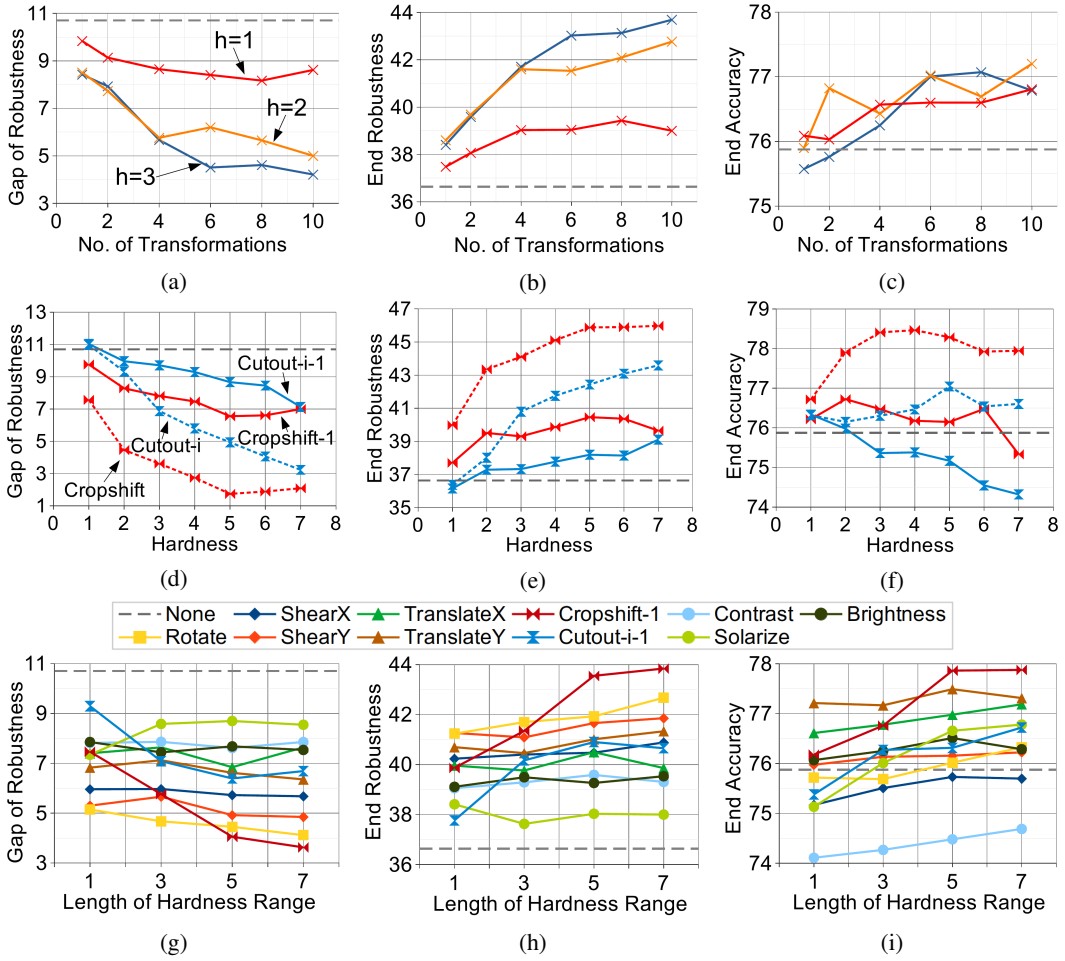

Figure 3: Performance of the models trained using augmentations with different type diversity (top row), spatial diversity (middle row) and strength diversity (bottom row). Spatial diversity rises from the restricted variant (solid lines) to the unrestricted variant (dashed lines) for the same transformation (line color) in Figs. 3d, 3e and 3f. Robustness is evaluated against PGD50.

standard training. Adversarial training can be considered as standard training plus gradient regularization (Li & Spratling, 2022). Roughly speaking, accuracy drops in this stage because the model's capacity is insufficient to fit the increasingly hard examples for the optimization of the clean loss (standard training) under the constraint of gradient regularization. Nevertheless, robustness could still increase due to the benefit of increasing robust generalization, i.e., smaller adversarial vulnerability. Third, accuracy (Fig. 2e) and robustness (Fig. 2f) drop together. Accuracy continues to decline due to the severer (standard) underfitting. Meanwhile, the harm of decreasing accuracy now outweighs the benefit of reduced robust overfitting, which results in the degeneration of robustness.

The graphs of Color, TranslateX and TranslateY are omitted from Figs. 2a, 2b and 2c because they exhibit exceptional behavior at some values of hardness. Nevertheless, these results generally show that robust overfitting is reduced and robustness is improved. These results are presented and discussed in Appendix D.1. Appendix D.2 provides a figure showing best accuracy as a function of hardness for individual augmentations. A more obvious downward trend with increasing hardness can be seen in this graph compared to the graph for end accuracy shown in Fig. 2b

## 3.2 DIVERSITY

To investigate the effects of augmentation diversity, variation in the augmentations was produced in three ways: (1) using varied types of transformations ("type diversity"); (2) varying the spatial area

to augment ("spatial diversity"); and (3) increasing the variance of the strength while keeping the mean strength constant ("strength diversity").

**Type diversity.** We uniformly drew $M$ transformations, with fixed and roughly equivalent hardness, from the same pool of transformations as the individual experiment in Section 3.1 but excluding Cutout and Cropshift. During training, one of these $M$ transformations was randomly sampled, with uniform probability, to be applied to each sample separately in a batch. Diversity increases as $M$ increases from 0 (no augmentation applied) to 10 (one augmentation from a pool of 10 applied). The experiments were repeated for three levels of hardness $\{1, 2, 3\}$. For all levels of hardness, the gap of robustness (Fig. 3a) reduces, and the end robustness (Fig. 3b) and accuracy (Fig. 3c) increases, as $M$ increases. These trends are more pronounced for higher hardness levels.

**Spatial diversity.** Transformations like Cutout and Cropshift (described in Section 4) have large inherent diversity due to the large number of possible crop and/or shift operations that can be performed on the same image. For example, there are 28x28 possible 4x4 pixel cutouts that could be taken from a 32x32 pixel image, all with the same strength of 4x4. In contrast, transformations like Shear and Rotation have only one, or if sign is counted, two variations at the same strength, and hence, have a much lower inherent diversity. To investigate the impact of this rich spatial diversity, we run the experiments to compare the performance of Cutout-i-1 with Cutout-i, and to compare Cropshift-1 and Cropshift, at various levels of hardness. In both cases the former variant of the augmentation method is less diverse than the latter. We observed that the rich spatial diversity in Cutout-i and Cropshift helps dramatically shrink the gap between the best and end robustness (Fig. 3d), and boost the end robustness (Fig. 3e) and accuracy (Fig. 3f) at virtually all hardness.

**Strength diversity.** Diversity in the strength was generated by defining four ranges of hardness: $\{4\}$, $\{3, 4, 5\}$, $\{2, 3, 4, 5, 6\}$ and $\{1, 2, 3, 4, 5, 6, 7\}$. During training each image was augmented using a strength uniformly sampled at random from the given range. Hence, for each range the hardness of the augmentation, on average, was the same, but the diversity of the augmentations increased with increasing length of the hardness range. Models trained with differing diversity were trained with each of the individual transformations defined in Section 3.1 excluding Color and Sharpness. Strength diversity for Cutout-1, Cropshift-1 and Rotate can be seen to significantly mitigate robust overfitting (Fig. 3g) and boost robustness (Fig. 3h), whereas for the other transformations it seems have no significant impact on these two metrics. Nevertheless, a clear increase in accuracy (Fig. 3i) is observed when increasing strength diversity for almost all transformations.

## 4 DIVERSE AND HARDNESS-BALANCED DATA AUGMENTATION

This section first describes Cropshift, our proposed version of Padcrop with enhanced diversity and disentangled hardness. Cropshift (Fig. 4; Algorithm 1) first randomly crops a region in the image and then shifts it around to a random location in the input space. The cropped region can be either square or rectangular. The strength of Cropshift is parameterized by the total number, $N$, of cropped rows and columns. For example, with strength $8$, Cropshift removes $l, r, t, b$ lines from the left, right, top and bottom borders respectively, such that $l + r + t + b = 8$. Cropshift significantly diversifies the augmented data in terms of both the content being cropped and the localization of the cropped content in the final input space. Furthermore, Cropshift offers a more fine-grained control on the hardness. In contrast, for Padcrop hardness is not directly related to the size of the padding, as for example, using 4 pixel padding can results in cropped images with a variety of total image content that is trimmed (from 4 rows and 4 columns trimmed, to no rows and no columns trimmed).

To mitigate robust overfitting, we propose a new data augmentation scheme with Improved Diversity and Balanced Hardness (IDBH). Inspired by Müller & Hutter (2021), we design the high-level framework of our augmentation as a 4-layer sequence: flip, crop, color/shape and dropout. Each has distinct semantic meaning and is applied with its own probability. Specifically, we implement flip using Horizontal Flip, crop using Cropshift, dropout using Random Erasing, and color/shape using a set of Color, Sharpness, Brightness, Contrast, Autocontrast, Equalize, Shear (X and Y) and Rotate. The color/shape layer, when applied to augment an image, first samples a transformation according to a probability distribution and then samples a strength from the transformation's strength range. This distribution and the strength range of each component transformation are all theoret-

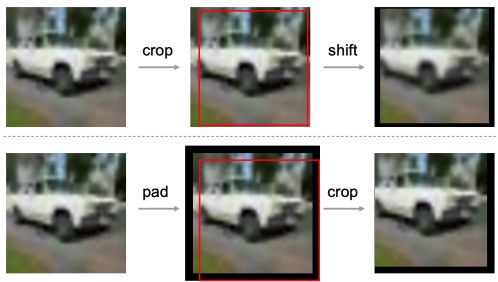

Figure 4: Illustration Cropshift (top) and Pad-crop (bottom) with equivalent hardness.

**Algorithm 1.** Pseudo code of Cropshift. Note: $randi(N)$ uniformly samples an integer between $[0, N)$.

```
w, h = get_image_size(img)
// crop region of orig.  image
cropx, cropy = randi(N), N − cropx
topx, topy = randi(cropx), randi(cropy)
w', h' = w − cropx, h − cropy
cropped = crop(img, topx, topy, w', h')
// shift the cropped region
x, y = randi(cropx), randi(cropy)
aug = zeros_like(img)
aug[x : x + w', y : y + h'] = cropped
```

ically available to optimize. Pseudo-code for the proposed augmentation procedure can be found in Appendix E.

The probability and the strength of each layer was jointly optimized by a heuristic search to maximize the robustness. It is important to optimize all layers together, rather than individually. First, this enables a more extensive and more fine-grained search for hardness so that a potentially better hardness balance can be attained. Moreover, it allows a certain hardness to be achieved with greater diversity. For example, raising hardness through Cropshift also improves diversity, while doing so through the color/shape layer hardly increases diversity. However, optimizing the parameters of all layers jointly adds significantly to the computational burden. To tackle this issue, the search space was reduced based on insights gained from the preliminary experiments and other work, and grid search was performed only over this smaller search space. Full details are given in Appendix E. A better augmentation schedule might be possible if, like AuA, a more advanced automatic search was applied. However, automatically searching data augmentation in adversarial training is extremely expensive, and was beyond the resources available to us.

IDBH improves diversity through the probabilistic multi-layered structure which results in a very diverse mixture of augmentations including individual transformations and their compositions. We further diversify our augmentation by replacing the conventional crop and dropout methods, Padcrop and Cutout, in AuA and TA with their diversity-enhanced variants Cropshift and Random Erasing respectively. IDBH enables balanced hardness, as the structure design and optimization strategy produce a much larger search space of hardness, so that we are able to find an augmentation that achieves a better trade-off between accuracy and robustness.

## 5 RESULTS

We adopt the following setup for training and evaluation (fuller details in Appendix C). The model architectures used were Wide ResNet34 with widening factor of 1 (WRN34-1), its 10x widened version WRN34-10 (Zagoruyko & Komodakis, 2016) and PreAct ResNet18 (PRN18) (He et al., 2016b). For PRN18, we report the result of two variants, weak and strong, of our method with slightly different parameters (hardness), because we observed a considerable degradation of best robustness on the strong variant when combined with SWA.

### 5.1 STATE-OF-THE-ART DATA AUGMENTATION FOR ADVERSARIAL TRAINING

From Tab. 1 it can be seen that our method, IDBH, achieves the state-of-the-art best and end robustness for data augmentations, and its improvement over the previous best method is significant. The robust performance is further boosted when combined with SWA. Moreover, our method is the only one on WRN34-1 that successfully improves the end robustness to be higher than the best robustness achieved by the baseline. On PRN18, IDBH[strong] improves the end robustness over the baseline's best robustness by $+1.78\%$, which is much larger than the existing best record ($+0.5\%$) achieved by AuA. This suggests that data augmentation alone, contrary to the previous failed attempts, can significantly beat the baseline augmentation with ES. More importantly, our method also presents the

Table 1: Performance of various data augmentation methods (w.o. SWA) and their weight-averaged variants (w. SWA) for WRN34-1 and PRN18 on CIFAR10. The best record is highlighted for each metric in each block. All methods were trained in the same way, except that Cutmix was trained longer for convergence as in Rebuffi et al. (2021). Robustness is evaluated against AA.

| Augmentation | w.o. SWA | | | | | | w. SWA | | | | | |
| | Accuracy (%) | | | Robustness (%) | | | Accuracy (%) | | | Robustness (%) | | |
| | best | end | diff. | best | end | diff. | best | end | diff. | best | end | diff. |
|---|---|---|---|---|---|---|---|---|---|---|---|---|
| Wide ResNet34-1 | | | | | | | | | | | | |
| baseline | 78.37 | 78.96 | -0.58 | 45.11 | 43.84 | 1.26 | 77.76 | 79.47 | -1.71 | 45.71 | 44.69 | 1.02 |
| Cutout | 77.65 | 78.41 | -0.76 | 45.22 | 44.43 | 0.78 | 76.86 | 79.09 | -2.23 | 45.74 | 45.37 | 0.37 |
| Cutmix | 74.12 | 75.52 | -1.40 | 45.10 | 44.49 | 0.61 | **77.95** | 78.06 | -0.11 | 45.27 | 45.32 | -0.05 |
| AuA | 74.59 | 74.93 | -0.34 | 42.62 | 43.28 | **-0.66** | 75.30 | 75.36 | **-0.06** | 43.44 | 43.52 | **-0.08** |
| TA | 73.19 | 73.37 | -0.18 | 42.06 | 41.92 | 0.14 | 73.41 | 73.53 | -0.12 | 42.79 | 42.74 | 0.06 |
| IDBH (ours) | **79.07** | **79.20** | **-0.13** | **46.15** | **45.65** | 0.50 | 77.82 | **79.83** | -2.01 | **46.70** | **46.26** | 0.44 |
| PreAct ResNet18 | | | | | | | | | | | | |
| baseline | 82.50 | 83.99 | -1.49 | 48.21 | 42.46 | 5.74 | 79.22 | 84.67 | -5.45 | 49.18 | 42.93 | 6.25 |
| Cutout | 83.35 | 84.14 | -0.79 | 49.18 | 47.75 | 1.43 | 81.98 | 84.37 | -2.39 | 50.18 | 48.76 | 1.42 |
| Cutmix | 82.47 | 81.51 | **0.96** | 49.73 | 48.09 | 1.65 | 84.09 | 85.70 | -1.62 | 50.57 | 47.50 | 3.07 |
| AuA | 83.41 | 84.04 | -0.62 | 49.15 | 48.71 | 0.44 | 82.08 | 84.20 | -2.12 | 49.53 | 49.59 | -0.07 |
| TA | 81.68 | 82.26 | -0.58 | 48.83 | 48.62 | **0.21** | 81.63 | 82.73 | **-1.11** | 49.25 | 49.42 | **-0.17** |
| IDBH[weak] (ours) | **84.98** | **85.82** | -0.84 | 50.34 | 48.94 | 1.40 | **84.18** | **86.45** | -2.27 | **51.73** | 49.88 | 1.85 |
| IDBH[strong] (ours) | 83.96 | 84.92 | -0.97 | **50.74** | **49.99** | 0.75 | 82.98 | 85.49 | -2.51 | 51.49 | **50.77** | 0.72 |

highest best and end accuracy on these architectures, both w. and w.o. SWA, except for WRN34-1 with SWA where our method is very close to the best. Overall, our method improves both accuracy and robustness achieving a much better trade-off between them.

Data augmentation was found to be sensitive to the capacity of the underlying model. As shown in Tab. 1, augmentations such as baseline, AuA and TA perform dramatically different on two architectures because they use the same configuration across the architectures. Meanwhile, augmentations like Cutout and ours achieve relatively better performance on both architectures but with different settings for hardness. For example, the optimal strength of Cutout is 8x8 on WRN34-1, but 20x20 on PRN18. Therefore, it is vital for augmentation design to allow optimization with a wide enough range of hardness in order to generalize across models with different capacity. A further discussion about generalization to alternative architectures like transformers is in Appendix F.1.

## 5.2 BENCHMARKING STATE-OF-THE-ART ROBUSTNESS WITHOUT EXTRA DATA

Tab. 2 shows the robustness of recent robust training and regularization approaches. IDBH matches the best robustness achieved by these methods on PRN18, and outperforms them considerably in terms of best robustness on WRN34-10. This is despite IDBH not being optimised for WRN34-10. In addition, our method also produces an end accuracy comparable to the best achieved by others suggesting a better trade-off between accuracy and robustness. More importantly, the robustness can be further improved by combining SWA and/or AWP with IDBH. This suggests that IDBH improves adversarial robustness in a way complementary to other regularization techniques. We highlight that our method when combined with both AWP and SWA achieves state-of-the-art robustness without extra data. We compare our method with those relying on extra data in Appendix D.3.

## 5.3 GENERALIZATION TO OTHER DATASETS

Our augmentation generalizes well to other datasets like SVHN and TIN (Tab. 3). It greatly reduces the severity of robust overfitting and improves both accuracy and robustness over the baseline augmentation on both datasets. The robustness on SVHN has been dramatically improved by +7.12% for best and +13.23% for end. The robustness improvement on TIN is less significant than that on the other datasets because we simply use the augmentation schedule of CIFAR10 without further optimization. A detailed comparison with those regularization methods on these two datasets can be found in Appendix D.4. Please refer to Appendix F.2 for generalization analysis to larger datasets like ImageNet (Deng et al., 2009) and Appendix C for the training and evaluation settings.

Table 2: Performance of methods for alleviating robust overfitting for PRN18 and WRN34-10 on CIFAR10. Robustness is evaluated against AA.

| Method | PreAct ResNet18 | | | | | | Wide ResNet34-10 | | | | | |
| | Accuracy (%) | | | Robustness (%) | | | Accuracy (%) | | | Robustness (%) | | |
| | best | end | diff. | best | end | diff. | best | end | diff. | best | end | diff. |
|---|---|---|---|---|---|---|---|---|---|---|---|---|
| AT (Madry et al., 2018) | 82.50 | 83.99 | -1.49 | 48.21 | 42.46 | 5.75 | 85.90 | 86.63 | -0.74 | 53.42 | 48.22 | 5.20 |
| TRADE (Zhang et al., 2019) | 81.19 | 82.48 | -1.29 | 49.03 | 46.80 | 2.23 | 84.65 | - | - | 53.08 | - | - |
| Pretraining (Hendrycks et al., 2019) | - | - | - | - | - | - | 87.89 | - | - | 54.92 | - | - |
| AWP (Wu et al., 2020) | 83.33 | 84.39 | -1.06 | 50.57 | 49.95 | 0.62 | 85.57 | - | - | 54.04 | - | - |
| MLCAT (Yu et al., 2022) | 84.10 | 84.77 | -0.67 | 50.70 | 50.32 | 0.38 | - | - | - | 54.65 | 54.56 | 0.09 |
| CONS (Tack et al., 2022) | **85.25** | **86.45** | -1.20 | 49.05 | 48.57 | 0.48 | **89.93** | 89.82 | **0.11** | 54.08 | 52.36 | 1.72 |
| KD (Chen et al., 2021) | 84.51 | 85.40 | -0.89 | 49.87 | 49.72 | 0.15 | 86.81 | 87.06 | -0.25 | 55.50* | 55.34* | 0.16 |
| TE (Dong et al., 2022) | 82.35 | 82.79 | -0.44 | 50.59 | 49.62 | 0.97 | - | - | - | - | - | - |
| IDBH[strong] (ours) | 83.96 | 84.92 | -0.97 | 50.74 | 49.99 | 0.75 | 88.61 | 89.12 | -0.52 | 55.83 | 54.01 | 1.82 |
| IDBH[weak]+SWA (ours) | 84.18 | **86.45** | -2.27 | 51.73 | 49.88 | 1.85 | 89.04 | **89.93** | -0.89 | 57.70 | 54.10 | 3.61 |
| IDBH[weak]+AWP (ours) | 82.98 | 83.03 | -0.05 | 52.27 | 52.21 | 0.06 | 88.47 | 88.94 | -0.47 | 57.88 | 57.68 | 0.20 |
| IDBH[weak]+AWP+SWA (ours) | 83.42 | 83.45 | **-0.03** | **52.46** | **52.52** | **-0.06** | 89.00 | 89.08 | -0.08 | **58.16** | **58.13** | **0.04** |

The source of the results in this table are described in Appendix C. Results marked * are evaluated against PGD20 so the AA robustness is expected to be lower.

Table 3: Performance of our method for PRN18 on SVHN and TIN. Robustness is evaluated by AA (AutoPGD) for SVHN (TIN).

| Data | Augmentation | Accuracy (%) | | Robustness (%) | |
| | | best | end | best | end |
|---|---|---|---|---|---|
| SVHN | baseline | 90.55 | 90.18 | 47.48 | 40.86 |
| | IDBH (ours) | **93.70** | **93.92** | **54.56** | **54.09** |
| TIN | baseline | 46.94 | 46.60 | 20.19 | 13.82 |
| | IDBH (ours) | **50.91** | **51.21** | **21.29** | **19.22** |

Table 4: Performance of variants of IDBH for PRN18 on CIFAR10. Robustness is evaluated by AA.

| SWA | Variant | Accuracy (%) | | Robustness (%) | |
| | | best | end | best | end |
|---|---|---|---|---|---|
| w.o. | padcrop | 83.74 | 84.82 | 50.15 | 49.25 |
| | cropshift | **83.96** | **84.92** | **50.74** | **49.99** |
| w. | padcrop | 83.11 | 85.21 | 51.14 | 49.40 |
| | cropshift | **84.18** | **86.45** | **51.73** | **49.88** |

## 5.4 ABLATION TEST

We find that Cropshift outperforms Padcrop in our augmentation framework. To compare them, we replaced Cropshift with Padcrop in IDBH and kept the remaining layers unchanged. The strength of Padcrop was then optimized for the best robustness separately for w. and w.o. SWA. As shown in Tab. 4, changing Cropshift to Padcrop in our augmentation observably degrades both accuracy and robustness both w. and w.o. SWA.

## 6 CONCLUSION

This work has investigated data augmentation as a solution to robust overfitting. We found that improving robust generalization for adversarial training requires data augmentation to be as diverse as possible while having appropriate hardness for the task and network architecture. The optimal hardness of data augmentation is very sensitive to the capacity of the model. To mitigate robust overfitting, we propose a new image transformation Cropshift and a new data augmentation scheme IDBH incorporating Cropshift. Cropshift significantly boosts the diversity and improves both accuracy and robustness compared to the conventional crop transformation. IDBH improves the diversity and allows the hardness to be better balanced compared to alternative augmentation methods. Empirically, IDBH achieves the state-of-the-art accuracy and robustness for data augmentations in adversarial training. This proves that, contrary to previous findings, data augmentation alone can significantly improve robustness and beat the robustness achieved with baseline augmentation and early-stopping. The limit of our work is that we did not have sufficient computational resources to perform more advanced, and more expensive, automatic augmentation search like AutoAugment, which implies that the final augmentation schedule we have described may be suboptimal. Nevertheless, the proposed augmentation method still significantly improves both accuracy and robustness compared to the previous best practice. We discuss the potential plans to improve in Appendix F.3.

## ACKNOWLEDGMENTS

The authors acknowledge the use of the research computing facility at King's College London, King's Computational Research, Engineering and Technology Environment (CREATE), and the Joint Academic Data science Endeavour (JADE) facility. This research was funded by the King's - China Scholarship Council (K-CSC).

**Reproducibility Statement.** Our methods including both Cropshift and IDBH can be easily implemented using the popular machine learning development frameworks like PyTorch (Paszke et al., 2019). These two algorithms Cropshift and IDBH are illustrated in pseudo codes in Algorithm 1 and Algorithm 2 respectively. The procedure of optimizing IDBH is described in detail in Appendix E. The full parameters of the optimal augmentation schedules we found are described in Tab. 12 and Tab. 11. The training and evaluation settings are described in Section 5 and Appendix C. To further facilitate the reproducibility, we are going to share our code and the pre-trained models with the reviewers and area chairs once the discussion forum is open, and will publish them alongside the paper if accepted.

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

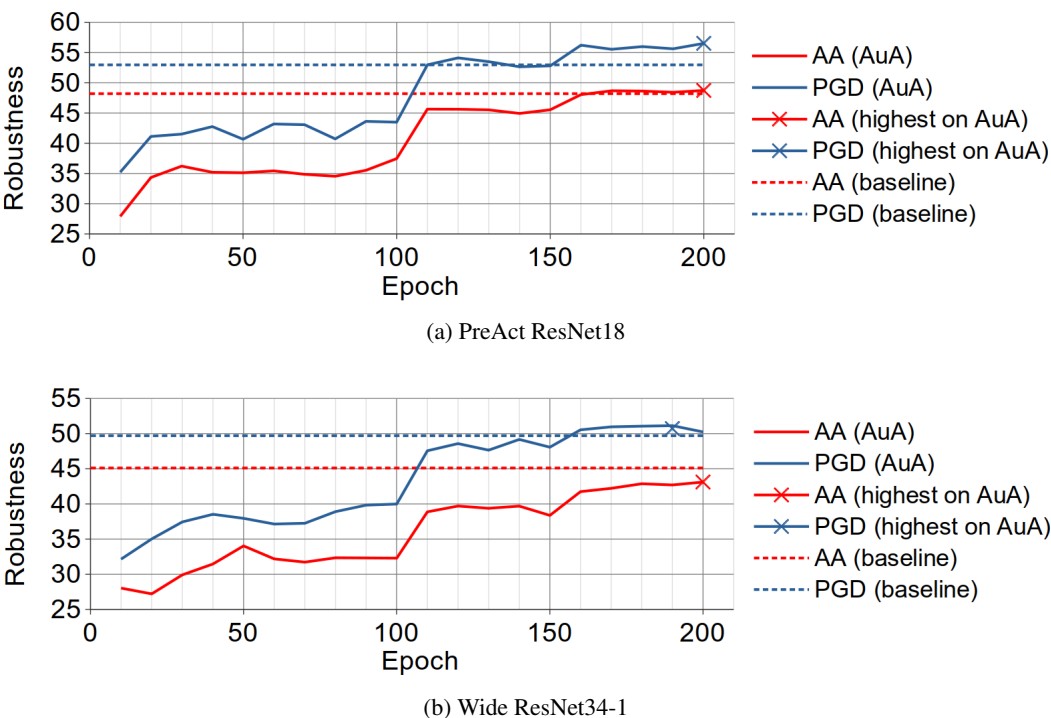

(a) PreAct ResNet18

(b) Wide ResNet34-1

Figure 5: The test robustness measured using PGD (blue lines) and AA (red lines) every 10 epochs during training. Results are shown, with solid lines, for PreAct ResNet18 (top) and Wide ResNet34-1 (bottom) trained with AutoAugment (AuA). The dashed lines represent the best robustness of the same models trained with the baseline data augmentation method, Flip-Padcrop, rather than AuA.

## A RECONCILING THE CONTRADICTORY RESULTS FOR AUTOAUGMENT

**Experimental Setting.** AutoAugment was implemented as in the original paper (Cubuk et al., 2019): Horizontal Flip in half chance, Padcrop with 4 pixels padding, Color/shape layer searched for CIFAR10, 16x16 Cutout. We adopted the implementation of the color/shape layer from PyTorch (Paszke et al., 2019). The model architectures were Wide ResNet34-1 and PreAct ResNet18. We trained the models using the same settings as Tack et al. (2022), which is actually our default training setting in Section 5. Robustness was evaluated against both Projected Gradient Descent (PGD) (Madry et al., 2018) with 10 steps and AutoAttack (AA) (Croce & Hein, 2020).

We observe, similar to Tack et al. (2022), that the PGD robustness of AuA-trained models increases over the baseline PGD robustness at the later stage of training for both network architectures (see Fig. 5. In contrast, the AA robustness is similar to (Fig. 5a), or is worse than (Fig. 5b), the baseline's. AA has been widely recognized as a more advanced attack that is better able to estimate adversarial robustness reliably (Croce & Hein, 2020). Therefore, we conclude that AuA fails to significantly boost the end robustness over the baseline, and the impressive improvement regarding PGD robustness is misleading. The misalignment between PGD robustness and AA robustness is more explicit in Fig. 5b, where the "best" checkpoint detected by PGD robustness is different from that of AA robustness. That's why the severity of robust overfitting is observed to be negative in Section 3.1 and Section 5.1.

Our results, and those of Tack et al. (2022), for AutoAugment on PreAct ResNet18 are inconsistent with previous work that found that the end robustness (Carmon et al., 2019), and even the best robustness (Gowal et al., 2021a), of AutoAugment is lower than the best robustness of the baseline augmentation. We suspect that these contradictory results may be partially due to inconsistent use of the term AutoAugment. Tack et al. (2022) explicitly state their AutoAugment includes Horizontal Flip, Padcrop and Cutout as in the original work (Cubuk et al., 2019). In contrast, Carmon et al. (2019); Gowal et al. (2021a) seem to refer AutoAugment as only the color/shape layer and

the remaining three component augmentations may or may not have been included. Specifically, Carmon et al. (2019) state AutoAugment is used in addition to Padcrop and Horizontal Flip, and Gowal et al. (2021a) state AutoAugment is used with the original setting, but no specific implementation information is given. Another possible account is that they used different training settings, and particularly a different capacity model, which has an important impact on the performance of data augmentations in adversarial training as shown in Secs. 3.1 and 5.1.

## B    DIFFERENCES BETWEEN DATA AUGMENTATION IN STANDARD AND ADVERSARIAL TRAINING

The idea of jointly handling hardness and diversity has been studied before in standard training (Gontijo-Lopes et al., 2021; Wang et al., 2021). However, the impact of hardness and diversity of data augmentation on adversarial training has never been researched before. This topic is particularly important because many previous attempts (discussed in Section 1) to solve robust overfitting using data augmentation methods from standard training failed and the cause of this failure is unclear. Our analysis provides some insight on why directly transferring data augmentation methods from standard training to adversarial training does not work.

More importantly, we find that the impact of hardness on adversarial training is very different from that on standard training. Gontijo-Lopes et al. (2021) and Wang et al. (2021) both claim that increasing both diversity and hardness will produce better data augmentation, which is in contrast to our finding that too hard data augmentation hurts both accuracy and robustness (we all share the same conclusion on diversity). This suggests that these two training paradigms, standard training and adversarial training, may require fundamentally different data augmentations. Therefore, we propose IDBH to maximize the diversity and balance the hardness according to the underlying model architecture, whereas Gontijo-Lopes et al. (2021); Wang et al. (2021) propose to maximize them both.

## C    EXPERIMENTAL SETTINGS

### C.1    TRAINING SETUP

**Section 3.** The experiments in this section were based on the following settings unless otherwise specified. The model's architecture was Wide ResNet34-1 (widening factor of 1) (Zagoruyko & Komodakis, 2016). The dataset was CIFAR10 (Krizhevsky, 2009). Data augmentation, if specified, was applied with 50% chance, i.e., half the time augmentation was applied and half the time it was not applied. Models were trained by stochastic gradient descent for 200 epochs with initial learning rate 0.1, divided by 10 at the epochs 100 and 150. The momentum was 0.9, the weight decay was 5e-4 and the batch size was 128. CrossEntropy loss was used. For both adversarial training and evaluation, we used the same attack, $l_\infty$ projected gradient descent (Madry et al., 2018), with a perturbation budget, $\epsilon$, of 8/255 and a step size of 2/255. The number of steps was 10 and 50 for training and evaluation respectively. Result were averaged over 5 runs. Experiments were run on Tesla V100 and A100 GPUs.

**Section 5.** The experimental settings were identical to those used in Section 3 unless specified below. The training method was PGD10 adversarial training (Madry et al., 2018). SWA was implemented as in Rebuffi et al. (2021). Robustness was evaluated against AA using the implementation of Kim (2021). In contrast to Section 3, data augmentation, if used, was always applied. We additionally evaluated on the datasets SVHN (Netzer et al., 2011) and Tiny ImageNet (TIN) (Le & Yang, 2015). As in previous work Yu et al. (2022), the baseline augmentation for SVHN was no data augmentation. The baseline data augmentation for TIN was the same as used for CIFAR10 i.e., Horizontal Flip (applied at half chance) and Padcrop with 4 pixel padding. Adversarial training was applied when training with TIN in exactly the same way it was as when training on CIFAR10. The reported results are averages over three runs, and the standard deviation is reported in Appendix D.5.

To train on SVHN, the initial learning rate was 0.01, the step size was 1/255 and the perturbation budget, $\epsilon$, was increased from 0 to 8/255 linearly in the first five epochs and then kept constants for the remaining epochs in order to stabilize the training, as suggested by Andriushchenko & Flam-

Table 5: The strength of individual transformations corresponding to the 7 degrees of hardness.

| | Hardness | | | | | | |
| --- | --- | --- | --- | --- | --- | --- | --- |
| | 1 | 2 | 3 | 4 | 5 | 6 | 7 |
| | Robustness(%) | | | | | | |
| | 45 | 40 | 35 | 30 | 25 | 20 | 15 |
| Transform | Strength | | | | | | |
| ShearX | 0.1 | 0.24 | 0.33 | 0.4 | 0.45 | 0.55 | 0.7 |
| ShearY | 0.1 | 0.2 | 0.3 | 0.37 | 0.45 | 0.55 | 0.7 |
| TranslateX | 1 | 3 | 4 | 6 | 7 | 9 | 11 |
| TranslateY | 1 | 3 | 4 | 6 | 7 | 9 | 10 |
| Rotate | 3 | 7 | 12 | 15 | 19 | 23 | 31 |
| Contrast | 0.92 | 0.82 | 0.73 | 0.67 | 0.62 | 0.56 | 0.5 |
| Brightness | 0.92 | 0.82 | 0.75 | 0.7 | 0.65 | 0.6 | 0.56 |
| Color | 0.7 | 0.3 | 0.1 | - | - | - | - |
| Sharpness | 0.7 | 0.3 | 0.01 | - | - | - | - |
| Solarize | 253 | 240 | 224 | 210 | 195 | 185 | 172 |
| Cropshift | 1 | 4 | 7 | 9 | 12 | 15 | 18 |
| Cutout-i | 3 | 6 | 9 | 11 | 13 | 15 | 18 |

marion (2020), otherwise, the same set-up was used as when training on CIFAR10. For adversarial evaluation, robustness on TIN was evaluated against AutoPGD (Croce & Hein, 2020) with 50 iterations and 5 restarts since we did not have access to sufficient computational resources to run AA for TIN on a PreAct ResNet18.

We re-optimized the strength of Cutout and Cutmix per model architecture. AuA was parameterized as in Cubuk et al. (2019) since we didn't have sufficient resource to optimize. TA is parameter-free so no tuning was needed. The size of cut-out area for Cutout was searched for within the range of {4x4, 6x6, 8x8, ..., 28x28}. The optimal size we found was 8x8 when useing Wide ResNet34-1 and 20x20 when using PreAct ResNet18. Following the procedure used in Yun et al. (2019), for Cutmix the value of the hyper-parameter $\alpha$ was searched for over the range {0.1, 0.25, 0.5, 1.0, 2.0, 4.0}. The optimal $\alpha$ we found was 0.1 on Wide ResNet34-1 and 0.25 on PreAct ResNet18. Cutout and Cutmix were applied with the default (baseline) augmentations in the order of Flip-Padcrop-Cutout and -Cutmix respectively. Similarly, for AuA (and TA) augmentions were applied in the order of Flip-Padcrop-AuA (TA)-Cutout as in Cubuk et al. (2019) (Müller & Hutter (2021)).

The source of the result being compared in Tab. 2 is as follows. For PreAct ResNet18, the result of AT, AWP and KD were determined by us. The result of TRADE is from Dong et al. (2022). For Wide ResNet34-10, the result of AT was determined by us. The result of TRADE, Pre-training and AWP is from Wu et al. (2020). Otherwise, the results are copied directly from the original work. All methods use the same training setting, except KD and Pre-training.

## C.2   STRENGTH AND HARDNESS OF INDIVIDUAL TRANSFORMATIONS

The 7 degrees of hardness used were: $\{1.04, 1.17, 1.34, 1.56, 1.87, 2.34, 3.12\}$. This corresponds to the denominator in Eq. (1), i.e., $Robustness(M, D'_{test})$, having values of $\{45, 40, 35, 30, 25, 20, 15\}\%$ as the PGD50 robustness of the model on the original test data (no data augmentation applied), i.e., $Robustness(M, D_{test})$, was 46.93%. The search range was constrained to be between 0 and 1 for the transformations Color, Sharpness, Brightness and Contrast. The corresponding strength of each individual transformation is described in Tab. 5. Note that the correspondence between strength and hardness is approximate because the real $Robustness(M, D'_{test})$ is not exactly equal to the nominal value given above, e.g., the $Robustness(M, D'_{test})$ of ShearX with strength 0.1 is only close to, instead of strictly equal to, 45%. Nevertheless, the variation between the real and the nominal $Robustness(M, D'_{test})$ for all transformations is small so this should not effect our analysis.

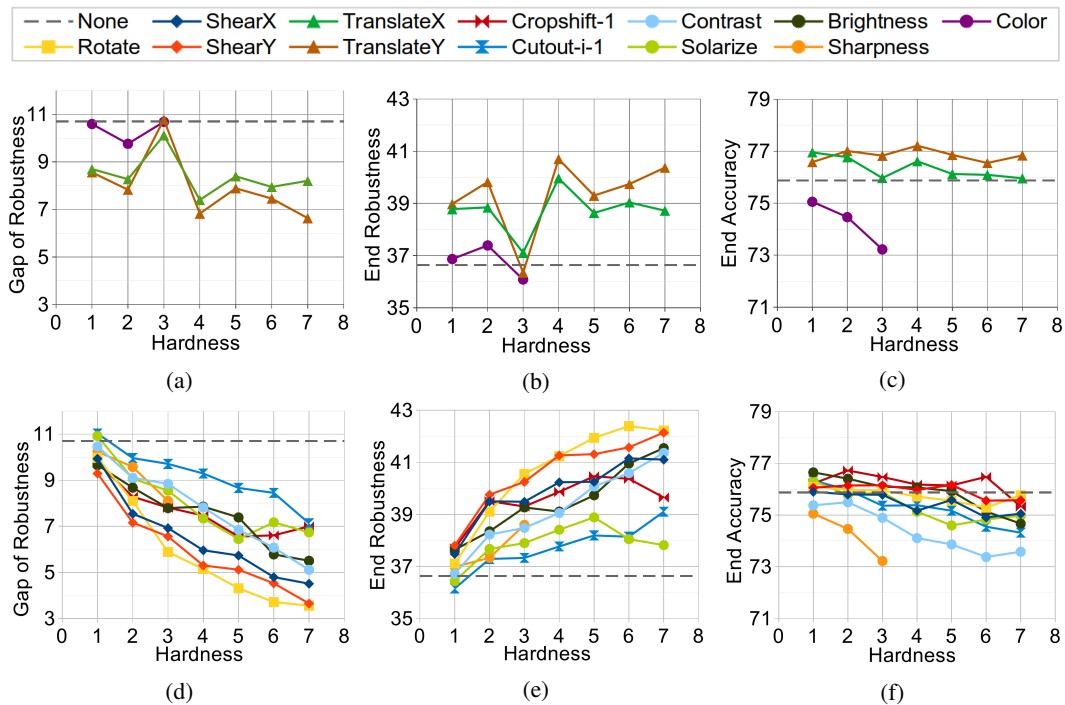

Figure 6: The comparison of the exceptional transformations Color, TranslateX and TranslateY (top row) and the others (bottom row) regarding the effect of hardness on the performance of adversarial training. The results at the bottom row also appear in Fig. 2 and are re-produced here to aid comparison. Robustness is evaluated against PGD50.

# D    ADDITIONAL EXPERIMENTAL RESULTS

## D.1    EXCEPTIONAL TRANSFORMATIONS IN HARDNESS EXPERIMENTS

The figures of Translate (X and Y) and Color present exceptional patterns, as shown in Fig. 6. First, robust overfitting and end robustness jumps abruptly, at hardness 3, to the same level of performance as training without augmentation (the gray dashed line). This suggests that training with any of them at this particular hardness does not provide a benefit, and can even impair, the robust generalization and end robustness. This is in contrast to the results produced with other values of hardness for these specific transformations, as well as the results for other augmentations. Hence, at some strengths these augmentation don't make the training data harder to fit although they, recalling how we measure the hardness, do make the test data more vulnerable to adversarial attacks. For Color, this may be due to a reduction in diversity as at hardness 3 (strength 0.1) this augmentation transforms colorful images into gray images (Fig. 7).

Second, ignoring the behavior at hardness 3 discussed above, for Translate, changing hardness does not produce the same clear trends in the three metrics that are seen with increasing hardness for the other transformations. This suggests that the complexity of data augmented by Translate doesn't increase consistently, at least within the evaluated range, with the increase in strength. A possible explanation for this is that the foreground object lies in the center of the image in most CIFAR10 samples and, therefore, translating the image will typically only remove background pixels, and introduce a black block at one border. A black block at the border is rare in the natural images, so adding this pattern to the data should increase the complexity. However, increasing the strength of Translate may have little further impact on the data complexity, because it only increases the size of the black regions while remove a few additional informative pixels from the opposite edge of the image. This contrasts with the other investigated transformations, like Shear, in which increased strength leads to increased distortion or, like Cropshift, which introduce more new patterns to the data (adding black blocks at more borders), with increasing strength. Hence, applying (versus not

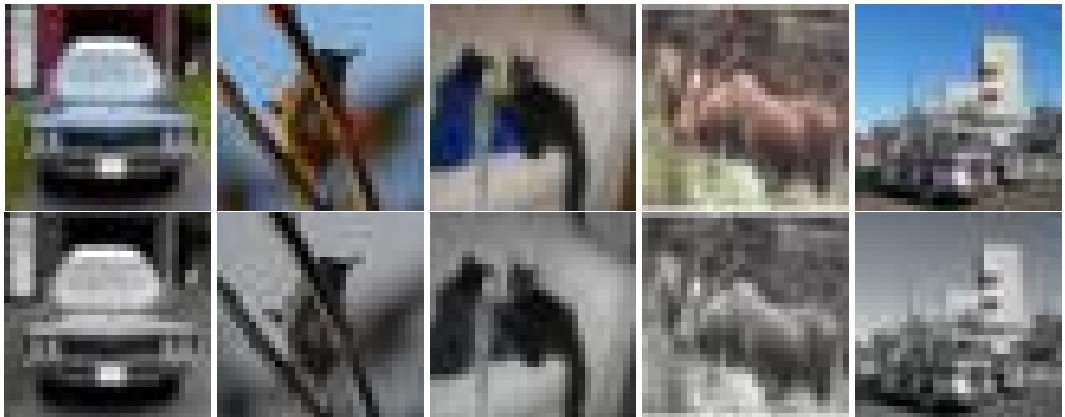

Figure 7: Illustration of the effect of Color transformation with hardness 3 (strength 0.1) on selected CIFAR10 images. The original images are in the top row, while the augmented images are in the bottom row.

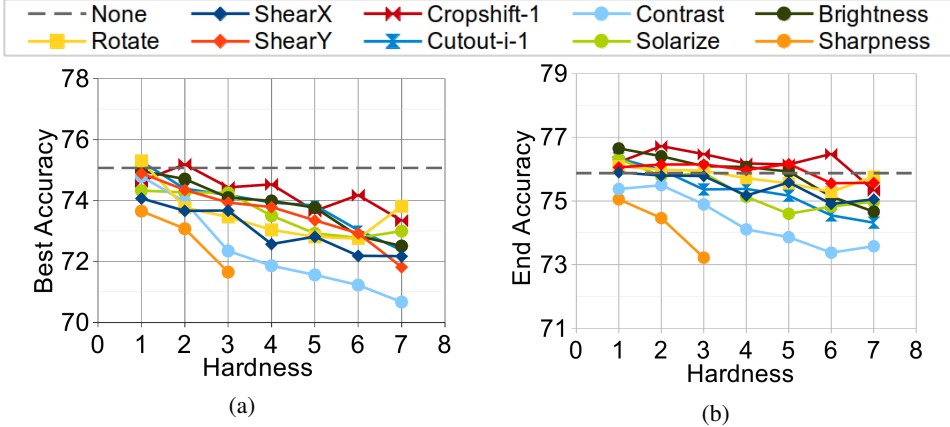

Figure 8: The effect of hardness of data augmentation on (a) best accuracy, and (b) end accuracy. The results for end accuracy also appear in Fig. 2b and are re-produced here to aid comparison.

applying) Translate effects the three metrics observably, but increasing the strength of Translate has little effect. Overall, these exceptions reflect a defect in measure of hardness, Eq. (1), and we leave the work of improving it to the future.

### D.2 FIGURES OF BEST ACCURACY FOR HARDNESS EXPERIMENTS

Fig. 8a shows the best accuracy with respect to the hardness for transformations excluding Translate and Color. It shows a more obvious downward trend for accuracy with increasing hardness, compared to the equivalent results for end accuracy Fig. 8b.

### D.3 COMPARISON WITH METHODS USING EXTRA DATA

Although our method fails to beat the robustness achieved by RST and PORT, it closes the gap between the performance of approaches that do not use additional training data with those that do use such additional data, as shown in Tab. 6. Our method doesn't require any additional data, whereas RST and PORT relies on a tremendous amount (0.5 and 10 millions) of unlabeled and synthetic data respectively. The acquisition of this volume of extra data is very expensive (Sehwag et al., 2022) and may be infeasible in some particular domains. Moreover, both RST and PORT mix the original and extra data in equal proportions in each mini-batch so that their actual batch size,

Table 6: Performance of the methods for improving adversarial training with and without extra data for Wide ResNet34-10 on CIFAR10. Robustness is evaluated against AA. Accuracy and Robustness are evaluated on the "best" checkpoint. The result of RST is copied from Wu et al. (2020).

| Method | Extra Data (Million) | Accuracy (%) | Robustness (%) |
|---|---|---|---|
| AT (Madry et al., 2018) | - | 85.90 | 53.42 |
| RST (Carmon et al., 2019) | 0.5 | **89.69** | 59.53 |
| PORT (Sehwag et al., 2022) | 10 | 87.00 | **60.60** |
| IDBH[strong] (ours) | - | 88.61 | 55.83 |
| IDBH[weak]+SWA (ours) | - | 89.04 | 57.70 |
| IDBH[weak]+AWP (ours) | - | 88.47 | 57.88 |
| IDBH[weak]+AWP+SWA (ours) | - | 89.00 | 58.16 |

Table 7: Performance of the methods for alleviating robust overfitting for PreAct ResNet18 on SVHN and TIN. Robustness is evaluated against AA for SVHN and and PGD20 for TIN. The results for the regularization methods are copied from the original works. Note that the original training set-up for TE (on SVHN) and KD (on TIN) is slightly different from ours.

| Dataset | Augmentation | Accuracy (%) | | | Robustness (%) | | |
|---|---|---|---|---|---|---|---|
| | | best | end | diff. | best | end | diff. |
| SVHN | baseline | 90.55 | 90.18 | **0.37** | 47.48 | 40.86 | 6.62 |
| | MLCAT | - | - | - | 51.90 | 49.76 | 2.14 |
| | TE | 90.09 | 90.91 | -0.82 | 51.44 | 50.61 | 0.83 |
| | ours | **93.70** | **93.92** | -0.22 | **54.56** | **54.09** | **0.47** |
| TIN | baseline | 46.94 | 46.60 | 0.34 | 23.05 | 14.41 | 8.64 |
| | CONS | 49.46 | 50.15 | **0.69** | 23.31 | 21.33 | 1.98 |
| | KD | 50.57 | **51.38** | -0.81 | 21.84 | **21.45** | **0.39** |
| | ours | **50.91** | 51.21 | -0.30 | **24.12** | 21.08 | 3.04 |

and hence computational cost, is twice that of methods, such as ours, that do not use additional data. The above drawbacks seriously limit the application of RST and PORT.

## D.4  COMPARISON WITH REGULARIZATION METHODS ON SVHN AND TIN

As shown in Tab. 7, IDBH achieves dramatic improvement of accuracy and robustness over those strong baselines on SVHN. Regarding TIN, IDBH, despite not being optimized for this particular dataset, achieves higher accuracy and robustness compared to KD and CONS.

## D.5  STANDARD DEVIATION DATA

Tabs. 8 and 9 provide the standard deviation data for the experimental result reported in Tab. 1, Tab. 2 and Tab. 3 respectively. Overall, the standard deviation of the robustness, both best and end, is no greater than 0.7.

## E  AUGMENTATION SCHEDULE AND OPTIMIZATION

Tab. 10 shows the reduced search space used to optimize hyperparameters for our proposed IDBH augmentation method (see Algorithm 2). The flip layer was fixed to Horizontal Flip at half chance following convention. For the crop layer, we searched over 8 combinations, where each had a different strength range (defined by a changed upper bound) and probability of applying the transformation. For Random Erasing, we considered two strength ranges, $(0.02, 0.33)$ and $(0.02, 0.5)$, and two probabilities of application, 0.5 and 1.0. In this case strength denotes the proportion of the image erased. The aspect ratio of the erased area was always uniformly sampled between 0.3 and

Table 8: Standard deviation of the experimental results in Tab. 1 and Tab. 2.

| Augmentation | w.o. SWA | | | | w. SWA | | | |
| --- | --- | --- | --- | --- | --- | --- | --- | --- |
| | Accuracy (%) | | Robustness (%) | | Accuracy (%) | | Robustness (%) | |
| | best | end | best | end | best | end | best | end |
| Wide ResNet 34-1 | | | | | | | | |
| baseline | 0.51 | 0.55 | 0.15 | 0.55 | 0.28 | 0.09 | 0.12 | 0.17 |
| Cutout | 0.18 | 0.43 | 0.11 | 0.31 | 0.11 | 0.15 | 0.09 | 0.15 |
| Cutmix | 1.57 | 1.28 | 0.29 | 0.32 | 0.15 | 0.08 | 0.11 | 0.20 |
| AuA | 1.35 | 0.52 | 0.28 | 0.12 | 0.41 | 0.30 | 0.49 | 0.36 |
| TA | 0.33 | 0.64 | 0.54 | 0.28 | 0.25 | 0.29 | 0.11 | 0.23 |
| IDBH (ours) | 0.35 | 0.39 | 0.06 | 0.12 | 0.22 | 0.18 | 0.10 | 0.04 |
| PreAct ResNet 18 | | | | | | | | |
| baseline | 0.31 | 0.53 | 0.57 | 0.23 | 0.22 | 0.13 | 0.11 | 0.16 |
| Cutout | 0.17 | 0.40 | 0.11 | 0.54 | 0.13 | 0.11 | 0.09 | 0.12 |
| Cutmix | 0.46 | 0.36 | 0.33 | 1.10 | 0.39 | 0.09 | 0.21 | 0.03 |
| AuA | 0.43 | 0.51 | 0.38 | 0.22 | 1.43 | 0.06 | 0.24 | 0.16 |
| TA | 0.31 | 0.20 | 0.32 | 0.40 | 1.83 | 0.16 | 0.39 | 0.26 |
| IDBH[weak] (ours) | 0.21 | 0.12 | 0.15 | 0.15 | 0.35 | 0.08 | 0.16 | 0.24 |
| IDBH[strong] (ours) | 0.42 | 0.04 | 0.06 | 0.25 | 0.13 | 0.23 | 0.25 | 0.22 |
| IDBH[weak]+AWP (ours) | 0.34 | 0.48 | 0.26 | 0.28 | 0.11 | 0.07 | 0.06 | 0.08 |
| Wide ResNet34-10 | | | | | | | | |
| baseline | 0.57 | 0.04 | 0.59 | 0.17 | 0.16 | 0.04 | 0.01 | 0.34 |
| IDBH[weak] (ours) | 1.58 | 0.33 | 0.70 | 0.50 | 0.08 | 0.16 | 0.11 | 0.18 |
| IDBH[strong] (ours) | 0.32 | 0.27 | 0.22 | 0.27 | 0.12 | 0.14 | 0.13 | 0.11 |
| IDBH[weak]+AWP (ours) | 0.45 | 0.27 | 0.23 | 0.25 | 0.14 | 0.11 | 0.30 | 0.20 |

Table 9: Standard deviation of the experimental results in Tab. 3.

| Dataset | Augmentation | Accuracy (%) | | Robustness (%) | |
| --- | --- | --- | --- | --- | --- |
| | | best | end | best | end |
| SVHN | baseline | 0.60 | 0.10 | 0.59 | 0.28 |
| | ours | 0.12 | 0.14 | 0.29 | 0.07 |
| TIN | baseline | 0.17 | 0.69 | 0.18 | 0.14 |
| | ours | 0.38 | 0.03 | 0.17 | 0.08 |

Table 10: The reduced search space used for each component of IDBH. Strength "1-X" means to uniformly sample a value between 1 and X, inclusive. Note that the strength of Cropshift is discrete so "1-5" is {1, 2, 3, 4, 5}. The strength of other component augmentations is continuous unless specified otherwise. "prob." denotes the probability of applying the transformation with the specified strength. For color/shape the strength is defined by a set of individual transformations as specified in Tab. 11.

| Horizontal Flip | | | Cropshift | | | Color/shape | | | Random Erasing | | |
|---|---|---|---|---|---|---|---|---|---|---|---|
| no. | strength | prob. | no. | strength | prob. | no. | strength | prob. | no. | strength | prob. |
| 1 | - | 0.5 | 1 | 1-5 | 0.833 (=5/6) | 1 | ColorBiased | 1 | 1 | - | 0 |
| | | | 2 | 1-6 | 0.857 (=6/7) | 2 | ShapeBiased | 1 | 2 | 0.02-0.33 | 0.5 |
| | | | 3 | 1-7 | 0.875 (=7/8) | | | | 3 | 0.02-0.5 | 0.5 |
| | | | 4 | 1-8 | 0.889 (=8/9) | | | | 4 | 0.02-0.33 | 1 |
| | | | 5 | 1-9 | 0.900 (=9/10) | | | | 5 | 0.02-0.5 | 1 |
| | | | 6 | 1-10 | 0.909 (=10/11) | | | | | | |
| | | | 7 | 1-11 | 0.917 (=11/12) | | | | | | |
| | | | 8 | 1-12 | 0.923 (=12/13) | | | | | | |

3.3. Apart from these 4 combinations ($2 \times 2$), we add one more case where Random Erasing is never applied, i.e. where the probability is zero.

Regarding the color/shape layer, two versions were implemented: a color transformations biased (ColorBiased) version, and a shape transformations biased (ShapeBiased) version, as defined in Tab. 11. These two realizations reflect the prior expectation that different datasets prefer essentially different types of transformation in standard training (Cubuk et al., 2019). The strength range of Color/Brightness/Contrast/Sharpness partially follows (Cubuk et al., 2019), but the lower bound of Brightness/Contrast is raised to 0.5 to be exempt from extremely hard (distorted) augmentations. Based on the results of the preliminary experiments, the strength range of Shear and Rotate was selected to have a relatively small average strength for the ColorBiased version, and a modest strength in the ShapeBiased version. The color (shape) transformations in the ColorBiased (ShapeBiased) instances are assigned a greater probability of being applied. The overall strength of this layer is relatively weak so that we can have more space to fine tune the strength of other layers particularly Cropshift which can boost the diversity alongside the hardness. We realize that there are many more possible implementations of the color/shape layer, considering additional implementations would greatly increase the computational burden of optimising the augmentation hyperparamters. Nevertheless, these two implementations reflect our intuition as to reasonable hyperparameters to search, and our results show that they are adequate to produce a large boost in accuracy and robustness. Overall, the total search space contains 80 ($8 \times 2 \times 5$) possible augmentation schedules.

We performed the grid search over the reduced search space for each model architecture on each dataset separately. The exception was we did not optimize our augmentation on TIN and Wide ResNet34-10 due to a lack of computational resources and simply used the same parameters as for PreAct ResNet18 on CIFAR10. The optimal schedules found are described in Tab. 12. We highlight that it is necessary to optimize the parameters for different architectures since our analysis (Section 3.1), as well as our search results (Tab. 12), suggests that the optimal data augmentation is very sensitive to the model architecture. We therefore expect that the results for Wide ResNet34-10 could be further improved if we optimize IDBH for it. The search procedure can be sped-up by being deployed on multiple GPUs in parallel. The efficiency can be further boosted by filtering out schedules that are likely to be worse based on the feedback of already evaluated schedules. For example, if a schedule appears to be overwhelmingly hard, degrading the accuracy and robustness, it is unnecessary to evaluate schedules that are harder.

Table 11: ColorBiased and ShapeBiased realizations of the color/shape layer. The strength and "prob." have the same interpretation as Tab. 10. $\pm[0.05\text{-}0.15]$ means a range of 0.05-0.15 with a randomly chosen sign. The strength value of Rotate was discrete. The probabilities of all component transformations in both realizations sums up to 1.

| Transformation | ColorBiased | | ShapeBiased | |
| --- | --- | --- | --- | --- |
| | strength | prob. | strength | prob. |
| Color | 0.1-1.9 | 0.125 | 0.1-1.9 | 0.08 |
| Brightness | 0.5-1.9 | 0.125 | 0.5-1.9 | 0.08 |
| Contrast | 0.5-1.9 | 0.125 | 0.5-1.9 | 0.04 |
| Sharpness | 0.1-1.9 | 0.125 | 0.1-1.9 | 0.08 |
| AutoContrast | - | 0.125 | - | 0.04 |
| Equalize | - | 0.125 | - | 0.08 |
| ShearX | $\pm[0.05\text{-}0.15]$ | 0.0625 | 0.05-0.35 | 0.15 |
| ShearY | $\pm[0.05\text{-}0.15]$ | 0.0625 | 0.05-0.35 | 0.15 |
| Rotate | $\pm[1\text{-}10]$ | 0.125 | 1-30 | 0.3 |

Table 12: The optimal augmentation schedules found for our method on various datasets and models.

| Data | Model | Variant | Horizontal Flip | | Cropshift | | Color/shape | | Random Erasing | |
| --- | --- | --- | --- | --- | --- | --- | --- | --- | --- | --- |
| | | | strength | prob. | strength | prob. | strength | prob. | strength | prob. |
| CIFAR10 | Wide ResNet34-1 | - | - | 0.5 | 1-5 | 0.833 | ColorBiased | 1 | - | 0 |
| CIFAR10 | PreAct ResNet18 | weak | - | 0.5 | 1-10 | 0.909 | ColorBiased | 1 | 0.02-0.33 | 0.5 |
| CIFAR10 | PreAct ResNet18 | strong | - | 0.5 | 1-10 | 0.909 | ColorBiased | 1 | 0.02-0.33 | 1 |
| SVHN | PreAct ResNet18 | - | - | 0 | 1-8 | 0.889 | ShapeBiased | 1 | 0.02-0.5 | 1 |

**Algorithm 2.** Pseudo-code for IDBH. $P_x$ is the probability of applying the transformation $x$. $S_y$ is the strength range of the transformation $y$. $rand()$ uniformly samples a floating-point number between 0 (inclusive) and 1 (exclusive).

**Function** `IDBH` (*img*) **:**
    **if** $rand() < P_{flip}$ **then**
         |   $img$ = horizontal_flip($img$)
    **end**

    **if** $rand() < P_{crop}$ **then**
         |   $s$ = uniform_sample($S_{crop}$)
         |   $img$ = cropshift($img$, $s$)
    **end**

    **if** $rand() < P_{color/shape}$ **then**
         |   /* taking an example of ColorBiased color/shape layer. */
         |   transform, $S_{transform}$ = sample_transform($ColorBiased$)
         |   $s$ = uniform_sample($S_{transform}$)
         |   $img$ = transform($img$, $s$)
    **end**

    **if** $rand() < P_{dropout}$ **then**
         |   $s$ = uniform_sample($S_{dropout}$)
         |   $img$ = erase($img$, $s$)
    **end**

    **return** $img$

## F ADDITIONAL DISCUSSION

### F.1 GENERALIZATION TO OTHER MODEL ARCHITECTURES

Vision transformer (ViT) (Han et al., 2023) is an emerging model architecture in the computer vision community. The latest ViT work (Zhai et al., 2022) achieved state-of-the-art clean accuracy on the ImageNet recognition challenge (Russakovsky et al., 2015). It is therefore naturally attractive to test if ViT could also excel regarding adversarial robustness. (Shao et al., 2023) observed that ViTs exhibited higher adversarial robustness compared to contemporary CNNs with standard training settings. (Mo et al., 2022) found that adversarial training was still necessary for ViTs to attain decent adversarial robustness. (Mo et al., 2022) then conducted extensive experiments to evaluate various adversarial training techniques for adversarial robustness. In one of the experiments, they found that CutMix (Yun et al., 2019) and Mixup (Zhang et al., 2018) were beneficial for adversarially-trained ViTs to obtain higher adversarial robustness, while RandAugment (Cubuk et al., 2020) impaired robustness instead. This is consistent to our observation that data augmentation methods like RandAugment designed for standard training can be too hard for adversarial training. In fact, (Mo et al., 2022) attributes the failure of RandAugment to the same cause: "RandAugment is too difficult for adversarial training of ViTs". Therefore, we believe that our method of balancing hardness for data augmentation should also improve the adversarial robustness of ViTs.

### F.2 GENERALIZATION TO LARGER DATASETS

We argue that Cropshift and IDBH can still be effective on larger datasets like ImageNet (Deng et al., 2009). The crop-based transformation RandomResizedCrop (RRC) has been widely used before in adversarial training on ImageNet (Wong et al., 2020; Shafahi et al., 2019). Although RRC is different from Cropshift's backbone crop transformation Padcrop, it is easy to adapt the idea of Cropshift to RRC to augment its diversity. For example, one can treat the outcome of RRC pipeline as the input image to the Cropshift pipeline. On the one hand, the way Cropshift augments data is unseen in RRC-augmented data so it creates effective new data instead of something duplicated. On the other hand, Cropshift retains most of the semantic information in the input images at normal strength, so the distribution shift caused by Cropshift can be controlled to be benign, i.e., learnable and beneficial. Regarding IDBH, it inherits component transformations and multi-layered structure from the precedent works AutoAugment (Cubuk et al., 2019) and TrivialAugment (Müller & Hutter, 2021), and optimizes the strength (hardness) of transformations for adversarial training. Given the superior performance of AutoAugment and TrivialAugment on ImageNet, we expect IDBH to improve adversarial training on ImageNet once appropriately optimized. Therefore, we believe that Cropshift and IDBH can benefit adversarial training on ImageNet. However, unfortunately, conducting adversarial training on ImageNet is beyond our computational resources so we can not provide empirical verification of our argument at this time.

### F.3 FUTURE WORKS FOR EFFICIENCY AND EFFECTIVENESS IMPROVEMENT

Low-efficiency heuristic search is a considerable drawback of our work that prevents it from scaling to larger datasets and being applied easily to new datasets. We believe that automatic search methods like FasterAA (Hataya et al., 2020) and DDAS (Liu et al., 2021) could provide a feasible and promising solution to this issue in the future. In general, they parameterize the underlying data augmentation sampler for gradient computation and then integrate it into the backpropagation pipeline so that the parameters of data augmentations can be optimized towards some predefined objectives, e.g., maximizing clean accuracy (Cubuk et al., 2019). Some automation techniques can be incorporated into the normal training pipeline to allow continuous, online, updating of the data augmentation strategy. This is in contrast to the offline data augmentation method adopted by our method where the data augmentation strategy is constant throughout training. Online data augmentation, ideally, should outperform the offline one, since it theoretically adapts data augmentation for models at every stage in training.

