# OpenReview forum: "Data augmentation alone can improve adversarial training"
_ICLR.cc/2023/Conference — ICLR 2023 poster_

### Official Review · Reviewer_4eAk · 2022-10-18

**Confidence:** 4
**Clarity, Quality, Novelty And Reproducibility:** See Strength And Weaknesses.
**Correctness:** 3
**Technical Novelty And Significance:** 2
**Empirical Novelty And Significance:** 3
**Recommendation:** 6

**Strength And Weaknesses:**

Strength
+ Compelling experimental results. The increase in accuracy and robustness is significant.
+ The proposed method would be easy to implement with (promised) available codes.

Weakness
+ Although it is clear that the hardness of a data augmentation method is measured by the robustness difference between the augmented model and the unaugmented one, it is unclear how it is controlled, e.g., to 7 different levels as the paper states. Is increasing hardness achieved by changing the strength of augmentation?
+ The idea of jointly handling hardness and diversity is not novel, which has been extensively studied in normal training [1]. Extending it to adversarial training seems incremental to me. Besides, the title looks close to [2].
+ It is claimed that “The probability and the strength of each layer (augmentation strategy) was jointly optimized by a heuristic search to maximize the robustness.” But the details of the heuristic search have not been illustrated. In this regard, it is unclear how the method optimizes the strength.

[1] AugMax: Adversarial Composition of Random Augmentations for Robust Training, NeurIPS 2021.

[2] Data augmentation can improve robustness, NeurIPS 2021.

**Summary Of The Paper:**

The paper proposes Cropshift, a new data augmentation to improve the robustness and accuracy of adversarial training. The design is based on the empirical findings that diversity matters and hardness can boost robustness at the cost of accuracy.

**Summary Of The Review:**

The method achieves great results, but the paper fails to illustrate important parts of the underlying mechanism.

---

> ### Author Response · Authors · 2022-11-14
> **Response Part 2**
>
> > Q3: It is claimed that “The probability and the strength of each layer (augmentation strategy) was jointly optimized by a heuristic search to maximize the robustness.” But the details of the heuristic search have not been illustrated. In this regard, it is unclear how the method optimizes the strength.
>
> **This is described in detail in Appendix D in the initial submission (Appendix E in the revised version of manuscript)**, as the page limit prevented us from including this information in the main text.
>
> Search over the entire hyperparameter space for our IDBH framework is beyond the resources we have, so **we start with reducing the search space to an affordable size**. Specifically, the flip layer was fixed to Horizontal Flip at half chance following convention. For the crop layer, we searched over 8 combinations, where each had a different strength range (defined by a changed upper bound) and probability of applying the transformation. For Random Erasing, we considered two strength ranges, $(0.02, 0.33)$ and $(0.02, 0.5)$, and two probabilities of application, 0.5 and 1.0. In this case strength denotes the proportion of the image erased. The aspect ratio of the erased area was always uniformly sampled between 0.3 and 3.3. Apart from these 4 combinations ($2\times2$), we add one more case where Random Erasing is never applied, i.e. where the probability is zero. Regarding the color/shape layer, two versions were implemented: a color transformations biased (ColorBiased) version, and a shape transformations biased (ShapeBiased) version. These two realizations reflect the prior expectation that different datasets prefer essentially different types of transformation in standard training [4]. **Overall, the total search space contains 80 ($8\times2\times5$) possible augmentation schedules*. **We performed the grid search over the reduced search space for each model architecture on each dataset separately**. The exception was we did not optimize our augmentation on TIN and Wide ResNet34-10 due to a lack of computational resources and simply used the same parameters as for PreAct ResNet18 on CIFAR10. For more details, please have a look at Appendix E in the revised manuscript.
>
> [4] Cubuk ea al. AutoAugment: Learning Augmentation Strategies From Data. CVPR 2019

---

> ### Author Response · Authors · 2022-11-14
> **Response Part 1**
>
> > Q1: Although it is clear that the hardness of a data augmentation method is measured by the robustness difference between the augmented model and the unaugmented one, it is unclear how it is controlled, e.g., to 7 different levels as the paper states. Is increasing hardness achieved by changing the strength of augmentation?
>
> Yes, **the hardness of a transformation is controlled by its strength**. We have described in detail how we define these 7 degrees of hardness and how we find the corresponding strength for each single transformation in Appendix B.2 in the initial submission (Appendix C.2 in the revised manuscript). We did not include them in the main text because we don't have sufficient space given the page limit.
>
> **The 7 degrees of hardness correspond to the denominator in Eq. 1, i.e., $Robustness(M, D_{test}')$, having values of ${45, 40, 35, 30, 25, 20, 15}%$** as the PGD50 robustness of the model on the original test data (no data augmentation applied), i.e., $Robustness(M, D_{test})$, was 46.93%. We then search for the strength to achieve the above robust accuracy for each transformation. For example, the strength of ShearX corresponding to 7 degrees of hardness is ${0.1, 0.24, 0.33, 0.4, 0.45, 0.55, 0.7}$. For complete details, please have a look at Appendix C.2 in the revised manuscript.
>
> > Q2: The idea of jointly handling hardness and diversity is not novel, which has been extensively studied in normal training [1]. Extending it to adversarial training seems incremental to me. Besides, the title looks close to [2].
>
> We agree with you that jointly handling hardness and diversity has been studied before in normal training. However, **the impact of hardness and diversity of data augmentation on adversarial training has never been researched before**. This topic is particularly important because many previous attempts (discussed in Sec. 1) to solve robust overfitting using data augmentation methods from normal training failed and the cause of this failure is unclear. **Our analysis provides some insight on why directly transferring data augmentation methods from normal training to adversarial training does not work**.
>
> More importantly, we find that **the impact of hardness on adversarial training is very different from that on normal training**. In both [1] and [2], they claim that increasing both diversity and hardness will produce better data augmentation, which is in contrast to our finding that too hard data augmentation hurts both accuracy and robustness (we all share the same conclusion on diversity). **This suggests that these two training paradigms, normal training and adversarial training, may require fundamentally different data augmentations.** Therefore, we propose IDBH to maximize the diversity and balance the hardness according to the underlying model architecture, whereas [1][2] proposes to maximize them both.
>
> The above text has been added to the Appendix B of the manuscript.
>
> **Regarding the title, we use it intentionally to highlight one of the key contributions of our work: data augmentation alone, i.e., even without any additional regularization, can significantly improve adversarial training regarding both accuracy and robustness.** This is in contrast to the previous findings including [3] that adversarial training can not be significantly improved by data augmentation alone.
>
> [1] Gontijo-Lopes et al. Tradeoffs in Data Augmentation: An Empirical Study. ICLR 2021
> [2] Wang et al. AugMax: Adversarial Composition of Random Augmentations for Robust Training. NeurIPS 2021
> [3] Rebuffi et al. Data Augmentation Can Improve Robustness. NeurIPS 2021

---

> > ### Comment · Reviewer_4eAk · 2022-11-17
> > **Thanks for the response**
> >
> > I would like to thank the authors for a detailed rebuttal, which addresses my issues in Q1 and Q3. However, it seems incorrect that "[1,2] claim increasing both diversity and hardness will produce better data augmentation", seeing "Affinity is a poor predictor of accuracy [1]" and "Diversity and hardness are two complementary dimensions of data augmentation to achieve robustness [2]". Intuitively, extremely hard examples (e.g., shifting out half of the image) would certainly decrease performance in both normal and adversarial training, and thus we always need proper hardness. As the authors admitted, "diversity is a panacea" is the same conclusion as normal training. I am suspicious of the claimed contributions that "the impact of hardness on adversarial training is very different from that on normal training".
> >
> > From my perspective, the contributions lie in the empirical success from tremendous efforts on the **heuristic trial of different augmentation combinations**. The grid search on the manual reduced search space strengthens such an impression. Is it possible to optimize augmentation strategies (and their parameters) by the gradients, which have already been obtained in adversarial training? Please see [a,b].
> >
> > [a] AugMax: Adversarial Composition of Random Augmentations for Robust Training, NeurIPS 2021.
> >
> > [b] DADA: Differentiable automatic data augmentation, ECCV 2020.

---

> > > ### Author Response · Authors · 2022-11-17
> > > **Response to the reply, part 2**
> > >
> > > We agree with you that one of our contributions is a successful data augmentation for adversarial training that is found by a heuristic search. However, **the more important contributions are the discovery of the effect of hardness and diversity for data augmentations in adversarial training, and how we use them to explain the failure of previous attempts (either less diversified or too hard) to solve robust overfitting using data augmentations and guide our design of Cropshift and IDBH to maximize diversity and maintain a wide range of hardness for search**. We would like to highlight that adversarial training is much more expensive compared to normal training so that searching over all possible augmentations in a brute-force or AutoAugment [6] way is prohibitively expensive to apply. Optimizing data augmentations for adversarial training is therefore challenging and less studied before. Our work can be concerned a first successful step to make data augmentation practical to use in adversarial training.
> > >
> > > Last, it is technically feasible to optimize data augmentation for adversarial training like [a, b] and the idea is interesting and valuable. The interesting thing is that **this technique can be used to generate online data augmentation strategies** that maintain a certain level of, or enhance, hardness relative to the underlying model (similar to the definition of hardness in [2]) throughout training. In contrast, the data augmentation strategy IDBH proposed in this work is offline and the data augmented by IDBH becomes easier to predict for the underlying model as it is trained more with IDBH. However, there are two concerns. First, **maximizing hardness or adversarial loss without any constraint may lead to the extremely hard augmentation** that, as suggested by our work, hurts the performance of adversarial training. Second, **the optimization algorithms used by [a, b] are sophisticated, but their performance may be suboptimal**. Specifically, [b] achieves accuracy comparable to AutoAugment [6] with dramatically reduced cost. However, [7] outperforms AutoAugment regarding accuracy on various datasets without any explicit search or optimization process. This implies that a sophisticated search or optimization process may be unnecessary.
> > >
> > > [6] Cubuk et al. AutoAugment: Learning Augmentation Strategies From Data. CVPR 2019
> > > [7] Müller et al. TrivialAugment: Tuning-Free Yet State-of-the-Art Data Augmentation. ICCV 2021

---

> > > > ### Comment · Reviewer_4eAk · 2022-12-08
> > > > **Thanks for the response**
> > > >
> > > > Thanks for the detailed clarification. I agreed with the authors that the hardness requirement of normal and adversarial training differs a lot, and this paper proposes a different definition of hardness. I thus raised my score to 6, and suggest the authors explicitly discuss the differences compared to existing work, and point out potential ways to study augmentation more efficiently (instead of manual work) in adversarial training.

---

> > > ### Author Response · Authors · 2022-11-17
> > > **Response to the reply, part 1**
> > >
> > > Many thanks for your reply.
> > >
> > > We agree with you that extremely hard examples would decrease performance in both normal and adversarial training. However, we don't think this idea was discovered or implemented in [1, 2]. When we say that "the impact of hardness on adversarial training is different from that on normal training", we mean that **it is different from the previous suggestions in [1, 2]** and **the degree of hardness that is beneficial for adversarial and normal training is very different**: augmentations like AutoAugment and TrivialAugment that have appropriate hardness for normal training are too hard for adversarial training as shown in Tab. 1 in our manuscript.
> > >
> > > [1] observed "For fixed values of Diversity, test accuracy generally increases with higher values of Affinity" and hence suggested
> > > "Furthermore, our work suggests that further improvements can be expected if one tries to increase both of the metrics, instead of just one of them as was the case in previous work". In our understanding, **they argue that increasing Affinity with decreasing Diversity may hurt accuracy, but boosts accuracy if Diversity is not reduced at the same time**. Note that [1] and our work adopt different measures of diversity in spite of similar Affinity (hardness) measure. In our opinion, [1] observed an increasing benefit of hardness because normal training is expected to have a larger tolerance on hardness compared to adversarial training so that the hardness of most evaluated augmentations is still within the beneficial range. **Overall, our conclusion on the effect of hardness is different from that in [1]**.
> > >
> > > [2] first randomly sample operators, including the strength of operators, like AuxMix [5] and then search for the worst-case mixing strategy of them. **Diversity here is inherited from AugMix style augmentation generation and hardness is maximized by an adversarial optimization pipeline on the mixing strategy.** We have not seen a similar expression like ours in [2] that hardness should be controlled to be moderate for the optimal performance. In our opinion, why maximizing hardness works in [2] may be because that the hardness is bounded by the randomly sampled augmentations and even the worst-case mixing of these augmentations is still within the beneficial range of hardness for standard training. Furthermore, **hardness in [2] is actually implicitly defined as the loss of the underlying model on augmented training data (i.e., the objective to be maximized in Eq. 4 in [2]), which is different from the definition of hardness in [1] and our work.** The essential difference is that hardness in [2] is computed against the underlying model being updated by training, whereas ours is computed against a pre-trained, constant, model that has never seen the evaluated augmentation in its training. **These two measures are distinct and can yield dramatically different hardness values for the same augmentation with the same strength.** Taking an example of shifting out half of the image, it may be concerned as easy if the model has been well trained on it (in spite of worse performance on test set), or hard if the model has never been trained on it, under the hardness measure of [2]. In contrast, it is always, likely, hard under our measure. From this perspective, [2] and our work investigated two essentially different types of hardness. **Overall, our work, compared to [2], studies a different type of hardness and discovers a different effect of hardness for data augmentation in the performance of a different training paradigm**.
> > >
> > > [5] Hendrycks et al. AugMix: A Simple Data Processing Method to Improve Robustness and Uncertainty. ICLR 2020

---

### Official Review · Reviewer_qfhi · 2022-10-24

**Confidence:** 4
**Correctness:** 3
**Technical Novelty And Significance:** 3
**Empirical Novelty And Significance:** 4
**Recommendation:** 6

**Clarity, Quality, Novelty And Reproducibility:**

Overall, the presentation remains to be improved. Although the proposed solution is not significantly novel, the findings presented in the analysis section are insightful to the community.

**Strength And Weaknesses:**

Strength
1. A novel finding that data augmentation alone could alleviate robust overfitting
2. This work also investigates when data augmentation help and proposes a promising data augmentation to improve model adversarial robustness

Weakness
1. IDBH is also comparable to Regularization methods in terms of improving model robustness. It is interesting to also study whether the proposed data augmentation can be combined with regularization to further improve model robustness.
2. The presentation remains to be improved. What is 'end accuracy'? When it can be lower/higher than 'best accuracy' in Tab. 1?
3. The main evaluation is conducted with PGD10. Does the PGD attack converge with 10 steps on the applied dataset and models?
4. In Tab. 3, the author aims to demonstrate that the proposed method works well across different datasets. However, some of the strong baselines in Table 2 are removed. Are there specific reasons behind this?


**Summary Of The Paper:**

The work demonstrates that data augmentation alone can alleviate robust overfitting.
The authors investigate what factors contribute to the robustness and point out that the hardness and diversity of the augmentation significantly influence the robustness and accuracy.
They propose a new image transformation method Cropshift and claimed to be more diverse compared to conventional methods.
Based on Cropshift, IDBH, a 4-layer image augmentation scheme is proposed and achieved SOTA performance and boosts a lot compared to previous data augmentation methods.


**Summary Of The Review:**

Given the significance of the insights provided in this work, I tend to weak accept this work.

---

> ### Author Response · Authors · 2022-11-14
> **Response Part 2**
>
> > Q3: The main evaluation is conducted with PGD10. Does the PGD attack converge with 10 steps on the applied dataset and models?}
>
> **The main evaluation of adversarial robustness is conducted with PGD50 for Sec. 3 and AutoAttack (AA) for Sec. 5. PGD10 is only used to find the "best'' checkpoint during training**. In the original version of the manuscript, PGD10 was also used to compare our method with Robust Self-Training (RST) in Tab. 2 of the original manuscript, since the only published results we found for RST on PreAct ResNet 18 were for PGD10 robustness. However, these results have now been removed, and our method is compared to RST using AA on Wide ResNet34-10 in the new table 6 because AA is widely accepted as more reliable attack than PGD.
>
> PGD attack normally doesn't converge with 10 steps on CIFAR10 and PreAct ResNet 18. That's why we mainly use PGD50 and AA for evaluation. PGD attack converges with 50 steps in our test. Note that PGD attack, even converged, is generally weaker than AA attack. Nevertheless, PGD is widely accepted as a quick evaluation due to its computational efficiency and validity of indicating the relative robustness (e.g. a network with higher PGD robustness usually has higher AA robustness if there is no gradient masking).
>
> > Q4: In Tab. 3, the author aims to demonstrate that the proposed method works well across different datasets. However, some of the strong baselines in Table 2 are removed. Are there specific reasons behind this?}
>
> We are sorry for the missing comparison. Those baselines are not included in Tab. 3 due to the page limit. Please find the comparison below:
>
> | Data | Method | Acc. Best |  Acc. End | Acc. Diff. | Rob. Best |  Rob. End | Rob. Diff. |
> |:----:|:------:|:---------:|:---------:|:----------:|:---------:|:---------:|:----------:|
> | SVHN | MLCAT  |     -     |     -     |      -     |   51.90   |   49.76   |    2.14    |
> | SVHN | TE     |   90.09   |   90.91   |    -0.82   |   51.44   |   50.61   |    0.83    |
> | SVHN | ours   | **93.70** | **93.92** |  **-0.22** | **54.56** | **54.09** |  **0.47**  |
> | TIN  | CONS   |   49.46   |   50.15   |  **0.69**  |   23.31   |   21.33   |    1.98    |
> | TIN  | KD     |   50.57   | **51.38** |    -0.81   |   21.84   | **21.45** |  **0.39**  |
> | TIN  | ours   | **50.91** |   51.21   |    -0.30   | **24.12** |   21.08   |    3.04    |
>
> We have revised the manuscript to include this comparison in Appendix D.4. As shown in the table, **IDBH achieves dramatic improvement of accuracy and robustness over those strong baselines on SVHN. Regarding TIN, IDBH, despite not being optimized for this particular dataset, achieves higher accuracy and robustness compared to KD and CONS**. The robustness, reported in the above table, is evaluated against AA on SVHN and against PGD20 on TIN. The original training setting of TE and KD is slightly different from ours.
>
> We realize that some baselines are still missing from the comparison in Tab. 2. This is because we can't find pre-trained models or results on the corresponding datasets for these methods in the existing literature, and we don't have sufficient resources to optimize these methods on new datasets. Fortunately, the data for two strongest baselines on CIFAR10, MLCAT and TE, is also available for SVHN.

---

> ### Author Response · Authors · 2022-11-14
> **Response Part 1**
>
> > Q1: IDBH is also comparable to Regularization methods in terms of improving model robustness. It is interesting to also study whether the proposed data augmentation can be combined with regularization to further improve model robustness.
>
> Thanks for your interest. This suggestion is interesting and valuable and has prompted us to test our data augmentation, IDBH, in combination with Adversarial Weight Perturbation (AWP) on CIFAR10 and PreAct ResNet-18 as shown below:
>
> |      Method      | Acc. Best |  Acc. End | Acc. Diff | Rob. Best |  Rob. End | Rob. Diff |
> |:----------------:|:---------:|:---------:|:---------:|:---------:|:---------:|:---------:|
> |     baseline     |   82.50   |   83.99   |   -1.49   |   48.21   |   42.46   |    5.75   |
> |  baseline + AWP  |   83.33   |   84.39   |   -1.06   |   50.57   |   49.95   |    0.62   |
> |       IDBH       | **84.98** |   85.82   |   -0.84   |   50.34   |   48.94   |    1.40   |
> |    IDBH + SWA    |   84.18   | **86.45** |   -2.27   |   51.73   |   49.88   |    1.85   |
> |    IDBH + AWP    |   82.98   |   83.03   |   -0.05   |   52.27   |   52.21   |    0.06   |
> | IDBH + AWP + SWA |   83.42   |   83.45   | **-0.03** | **52.46** | **52.52** | **-0.06** |
>
> IDBH when combined with AWP achieves a significantly higher robustness. The record can be further improved by combining Stochastic Weight Averaging (SWA) as well. This suggests that **IDBH improves adversarial robustness in a way complementary to other regularization techniques**. We have revised the manuscript to include this comparison on PreAct ResNet18 and Wide ResNet34-10 in Tab.2 in the revised manuscript.
>
> We adopt the hyperparameters of AWP as the original setting [1]. IDBH here refers to the weak version reported in our paper, since the strong version with AWP appears over-regularized. In fact, even the weak version of IDBH also appears a little bit over-regularized. A higher robustness and accuracy is thus foreseen if we re-optimize (reduce the strength of) IDBH with AWP.
>
> [1] Wu et al. Adversarial Weight Perturbation Helps Robust Generalization. NeurIPS 2020
>
> > Q2: The presentation remains to be improved. What is 'end accuracy'? When it can be lower/higher than 'best accuracy' in Tab. 1?
>
> We are sorry that our presentation confused you. This term is explained in the first paragraph of Sec. 3.
> **End (best) accuracy/robustness refers to the accuracy/robustness of the last (best) checkpoint during training**. Checkpoints are saved at the end of each epoch. The end checkpoint is the checkpoint saved at the end of training. The best checkpoint is the checkpoint with the highest robust accuracy against PGD10 on the test set.
>
> **End accuracy is usually higher than best accuracy**. This is because the model overfits the training adversary, so as the implicit regularization effect of adversarial training weakens, as training progresses. The clean accuracy, hence, increases with the decrease in adversarial robustness at the later stage of training. However, this is not always the case. For example, adversarial training with Cutmix on PreAct ResNet-18 (Tab. 1) shows that end accuracy can be lower than best accuracy. This may be caused by Cutmix generating unrealistic images and employing implicit label smoothing, in contrast to other evaluated data augmentations. The general reason why clean accuracy sometimes degrades alongside robustness in robust overfitting is less studied and remains unclear in the community. In our opinion, this phenomenon can be investigated as an independent research question and is beyond the scope of the current work.

---

### Official Review · Reviewer_5354 · 2022-10-24

**Confidence:** 4
**Clarity, Quality, Novelty And Reproducibility:** The paper is clear, novel and seems p…
**Correctness:** 4
**Technical Novelty And Significance:** 3
**Empirical Novelty And Significance:** 3
**Recommendation:** 6

**Strength And Weaknesses:**

Strength:
1) Claims are well motivated with a thorough study based on hardness and diversity. The story is quite clear with data augmentations which should be adapted to the capacity of the model.
2) The introduced data augmentation is tailored for robustness and gives indeed a significant boost in performance.
3) Thorough experiments with many ablation studies.

Weakness:
1) Cropshift works well with CIFAR, SVHN and TinyImageNet but datasets with larger images such as ImageNet typically benefit from other types of data augmentations than small datasets. So it is not sure that Cropshift would be helpful in that case.

**Summary Of The Paper:**

This paper studies adversarial training in the context of image classification. The authors show that data augmentation alone can improve robustness. In the previous literature, works showed that augmentation needed to be combined with other components in order to improve performance. The authors propose a new crop transformation called Cropshift which improve robust performance.

**Summary Of The Review:**

The idea of proposing a new data augmentation specially designed for robustness is novel and an interesting research direction. The authors provide extensive experiments to support their claims and give an interesting analysis frame with the hardness and diversity study.

---

> ### Author Response · Authors · 2022-11-14
> **Response**
>
> > Q1: Cropshift works well with CIFAR, SVHN and TinyImageNet but datasets with larger images such as ImageNet typically benefit from other types of data augmentations than small datasets. So it is not sure that Cropshift would be helpful in that case.
>
> We agree with you that ImageNet may benefit from other types of data augmentations than small datasets. In fact, even for small datasets themselves, CIFAR10 and SVHN benefit from different types of data augmentation: CIFAR10 prefers color-biased transformations like Contrast, while SVHN prefers shape-biased transformations like Shear [1]. ImageNet prefers, according to [1], color-biased transformations and Rotate for standard training. If this is the different type of data augmentation you are referring to, our data augmentation framework IDBH can support this preference through the color/shape layer. If not, IDBH can still incorporate new transformations by adding new layers for them. **It is therefore unnecessary to discard Cropshift to include other types of data augmentations in IDBH as long as there is no conflict between them**.
>
> If you are questioning the effectiveness of Cropshift on ImageNet we argue it is helpful. **The crop-based transformation RandomResizedCrop (RRC) [2] has been widely used before in adversarial training on ImageNet [3][4]**. Although RRC is different from Cropshift's backbone crop transformation Padcrop, **it is easy to adapt the idea of Cropshift to RRC to augment its diversity**. For example, one can treat the outcome of RRC pipeline as the input image to the Cropshift pipeline. On the one hand, the way Cropshift augments data is unseen in RRC-augmented data so it creates effective new data instead of something duplicated. On the other hand, Cropshift retains most semantics in the input images under the normal strength, so the distribution shift caused by Cropshift can be controlled to be benign, i.e., learnable and beneficial. Therefore, we believe that Cropshift can also benefit adversarial training on ImageNet. We are sorry that conducting adversarial training on ImageNet is beyond our computational resources so we can't provide empirical verification for our argument at this time.
>
> [1] Cubuk ea al. AutoAugment: Learning Augmentation Strategies From Data. CVPR 2019
> [2] RandomResizedCrop: https://pytorch.org/vision/stable/generated/torchvision.transforms.RandomResizedCrop.html\#torchvision.transforms.RandomResizedCrop
> [3] Wong et al. Fast is better than free: Revisiting adversarial training. ICLR 2020
> [4] Shafahi et al. Adversarial Training for Free!. NeurIPS 2019

---

> > ### Comment · Reviewer_5354 · 2022-11-29
> > **Thank you for the rebuttal**
> >
> > I thank the authors for the rebuttal which clarifies my interrogations so I will keep my rating.

---

### Official Review · Reviewer_KVU6 · 2022-10-27

**Confidence:** 4
**Correctness:** 3
**Technical Novelty And Significance:** 3
**Empirical Novelty And Significance:** 3
**Recommendation:** 6

**Clarity, Quality, Novelty And Reproducibility:**

The paper is clear and well-written. The novel aspects are - understanding the impact of hardness and diversity of augmentations in adversarial training, designing Cropshift for better diversity, and the IDBH augmentation pipeline.

**Strength And Weaknesses:**

Strengths -
- The analysis on the impact of increasing hardness and diversity of augmentations is insightful.
- The proposed method shows good improvements on PreActResNet-18 when compared to baselines.
- The authors perform a more robust evaluation of prior work by Tack et al., and show that their claim of superior robustness using AutoAugment is incorrect.

Weaknesses -
- As the authors mention, the selection of augmentations is currently a bottleneck, which makes it impractical to use, due to the requirement of manually finding the best set of augmentations. Even using strategies like the one used in AutoAugment is expensive due to the large search space.
- WideResNet-34 results seem to be much lower than the expected results [1]. Moreover, it is unclear why this is even lower than the results of PreAct-ResNet-18.
- It is unclear why increasing hardness degrades clean accuracy and improves robust accuracy. If the model capacity is insufficient for learning benign samples, it should be harder to learn adversarial samples. Moreover, the argument of robust generalization can also hold for standard generalization.

Suggestions -
-  This seems a bit confusing and different from the algorithm - "Cropshift first randomly crops a region in the image and then shifts it around to a random location in the input space." - could be rephrased.


[1] Pang et al., Bag of Tricks for Adversarial Training

**Summary Of The Paper:**

This work aims to address the robust overfitting issue in adversarial training by using data augmentations. Prior works have shown that adversarial training does not benefit from augmentations such as AutoAugment. The authors firstly study the role of hardness and diversity of augmentations in robustness and accuracy, and find that diversity improves both, while increasing hardness can improve robustness at the cost of clean accuracy initially and later degrades both. To improve diversity, the authors propose a new augmentation Cropshift, and further use this in the proposed augmentation pipeline - Improved Diversity and Balanced Hardness (IDBH) which has better diversity and well-balanced hardness.


**Summary Of The Review:**

Although there are several novel and interesting aspects in the paper as mentioned above, the results of baselines and proposed approach are suboptimal for WideResNet-34 when compared to prior works.

**Post-Rebuttal comments:** I thank the authors for their response which addresses many of my concerns. I suggest the authors include the explanation of - "why increasing hardness degrades clean accuracy and improves robust accuracy" in the paper as well, specifically the clarification that, "model capacity is insufficient to fit benign samples under a stronger regularization from adversarial training caused by increasing hardness".

Clarification on the suggestion -  Based on my understanding of the algorithm, the explanation can be updated to "Cropshift first randomly crops a region in the image and then places it at a random location on a new blank image."

---

> ### Author Response · Authors · 2022-11-14
> **Response Part 2**
>
> > Q3: It is unclear why increasing hardness degrades clean accuracy and improves robust accuracy. If the model capacity is insufficient for learning benign samples, it should be harder to learn adversarial samples. Moreover, the argument of robust generalization can also hold for standard generalization.
>
> Generally speaking, **robust accuracy can increase with the decrease in clean accuracy if adversarial vulnerability decreases more than clean accuracy**. We argue that **increasing hardness of data augmentation enhances the implicit regularization effect of adversarial training so that the underlying model's adversarial vulnerability is reduced**. To illustrate, we approximate adversarial loss using Taylor's theorem following [2] as:
> \begin{equation}
> \mathcal{L}(x+\delta) \approx \mathcal{L}(x) + \sum_i^d \nabla_{x_i}\mathcal{L}(x)\delta_i
> \end{equation}
> where $\nabla_{x_i}\mathcal{L}(x)$ is the first-order gradient of loss $\mathcal{L}(x)$ w.r.t. the input variable $x_i$ ($d$ is the number of dimensions of $x$) and $\delta_i$ is the adversarial perturbation added to $x_i$.
> From this perspective, adversarial training can be concerned as standard training with gradient regularization.
> Robust accuracy increases with an increase in clean accuracy or a decrease in the magnitude of the loss gradients. Therefore, robust accuracy may increase with the decrease in clean accuracy if the magnitude of loss gradients is reduced.
>
> As adversarial training overfits, the gradients on the training data continue to decrease so that the training adversarial loss drops. Meanwhile, the gradients on the unseen data grow up so that the test adversarial loss increases. Increasing hardness of data augmentation creates new unseen data in the training set and, by training on them, forces the loss gradients to be small over a larger region. The regularization effect of adversarial training is thus enhanced. Increasing hardness, compared to increasing diversity, generates new data that is farther away from the original data so that the regularization effect of adversarial training is enhanced more. The model capacity becomes less sufficient to fit benign samples with the increasing regularization as hardness increases. After reaching a certain limit of regularization strength, the model begins to underfit benign samples and the clean accuracy drops with the increase of hardness and regularization strength. Note that adversarial vulnerability (loss gradients) is also reduced with the increase of hardness, and robust accuracy may increase if adversarial vulnerability drops more than clean accuracy. **When we state in the paper that model capacity is insufficient to fit benign samples, we mean that model capacity is insufficient to fit benign samples under a stronger regularization from adversarial training caused by increasing hardness**.
>
> [2] Wang et al. Bilateral Adversarial Training: Towards Fast Training of More Robust Models Against Adversarial Attacks. ICCV 2019
>
> > Suggestion 1: This seems a bit confusing and different from the algorithm - "Cropshift first randomly crops a region in the image and then shifts it around to a random location in the input space." - could be rephrased.
>
> We are uncertain as to why the reviewer finds this statement is confusing, or how it differs from the definition of Cropshift given in the algorithm, and would therefore appreciate more guidance on this issue. We have revised the algorithm defining Cropshift in the manuscript to make it easier to follow.

---

> ### Author Response · Authors · 2022-11-14
> **Response Part 1**
>
> > Q1: As the authors mention, the selection of augmentations is currently a bottleneck, which makes it impractical to use, due to the requirement of manually finding the best set of augmentations. Even using strategies like the one used in AutoAugment is expensive due to the large search space.
>
> We agree with you that how to find/design proper data augmentation for adversarial training (AT) is an important yet challenging problem. It's challenging because AT typically costs about 10 times more computation than standard training (ST). This makes the popular automatic data augmentation search technique like AutoAugment prohibitively expensive to apply. Nevertheless, **our result suggests that a proper data augmentation can significantly boost both accuracy and robustness for AT**. This proves that data augmentation is a promising and perhaps vital, like in ST, technique for solving overfitting in AT.
>
> To tackle the selection issue, our first contribution is **the discovery of two principles for selecting the optimal data augmentation for AT**. Optimal data augmentation should have (1) maximal diversity and (2) appropriate hardness (dependent upon model capacity). We then propose a new image transformation Cropshift and a new augmentation framework IDBH (based on Cropshift) to implement the above principles. Last, **we demonstrate that a significant performance boost can be attained by an affordable heuristic search on our IDBH framework for the CIFAR10 and SVHN datasets**. The search space is reduced to 80 augmentation schedules based on the insight from the previous work and the preliminary experiments. Therefore, our work can be considered a first step towards making data augmentation practical to use for adversarial training, but we agree that there is still much scope for further improvement.
>
> > Q2: WideResNet-34 results seem to be much lower than the expected results [1]. Moreover, it is unclear why this is even lower than the results of PreAct-ResNet-18.
>
> Our result is lower than your expectation because **the term "Wide ResNet34" in the original manuscript refers to Wide ResNet34-1 (widening factor 1) instead of the commonly used Wide ResNet34-10 (widening factor 10)**. A detailed comparison between these two versions of Wide ResNet34 can be found in [1]. Our result of Wide ResNet34-1 is consistent to the result reported by [1]. We have revised the manuscript to explicitly show the widening factor of Wide ResNet used.
>
> Wide ResNet34-1 was used to show how accuracy and robustness are compromised when data augmentation is overly hard for the underlying small-capacity model (see the result of AuA and TA in Tab. 1 of the manuscript). We did not report the result on Wide ResNet34-10 in the initial submission but we have now produced some result for it that are shown below:
>
> |       Method      | Acc. Best |  Acc. End | Acc. Diff. | Rob. Best |  Rob. End | Rob. Diff. |
> |:-----------------:|:---------:|:---------:|:----------:|:---------:|:---------:|:----------:|
> | baseline          |   85.90   |   86.63   |    -0.74   |   53.42   |   48.22   |    5.20    |
> | IDBH-weak         |   87.03   |   89.48   |    -2.44   |   54.16   |   52.93   |    1.23    |
> | IDBH-strong       |   88.61   |   89.12   |  **-0.52** |   55.83   |   54.01   |    1.82    |
> | baseline + SWA    |   84.51   |   86.67   |    -2.16   |   54.68   |   48.55   |    6.13    |
> | IDBH-weak + SWA   | **89.04** | **89.93** |    -0.89   | **57.70** |   54.10   |    3.61    |
> | IDBH-strong + SWA |   88.12   |   89.76   |    -1.65   |   57.29   | **55.46** |    1.83    |
> | AWP               |   85.57   |     -     |      -     |   54.04   |     -     |      -     |
> | MLCAT             |     -     |     -     |      -     |   54.65   |   54.56   |  **0.09**  |
>
> We observe that IDBH consistently improves upon the baseline regarding both accuracy and robustness both with and without SWA applied. We have also included the result of AWP and MLCAT from their original papers in the table for comparison. IDBH[strong] alone boosts both accuracy and robustness, and even more significantly when combined with SWA, over these two regularization methods. **This confirms that the proposed approach is superior, instead of suboptimal, for Wide ResNet34-10 when compared with prior works**. Note that IDBH reported here is still the same one as used on PreAct ResNet-18 and we believe a better result can be attained by optimizing it for Wide ResNet34-10.
>
> We have revised the manuscript to include the result of Wide ResNet34-10 in Tab. 2.
>
> [1] Wu et al. Do Wider Neural Networks Really Help Adversarial Robustness? NeurIPS2021

---

### Author Response · Authors · 2022-11-14
**Summary of Revision**

Dear Reviewers,

We much appreciate the time and effort that you have dedicated to reviewing and providing your valuable feedback on our manuscript. We have revised our manuscript in light of your suggestions and the main changes are summarized here:
1. the result of our method on Wide ResNet34-10 is added to Tab. 2 as suggested by reviewer KVU6
2. more robust training and regularization methods are compared in Tab. 2
3. the result of combining our method with the regularization method AWP is added to Tab. 2 as suggested by reviewer qfhi
4. A comparison with regularization methods on SVHN and TIN is added at Appendix D.4 as suggested by reviewer qfhi
5. A detailed comparison with those methods relying on extra data is added at Appendix D.3

We would like to highlight two new records we achieved in new experiments. **Our method achieves 52.52% robustness (evaluated against AA) for PreAct ResNet18 and 58.16% for Wide ResNet34-10 on CIFAR10.** These, to our best knowledge, benchmark state-of-the-art robustness on these two architectures in the setting without extra data used.

We have also uploaded our code with the submission and will publish the code and pre-trained models on Github once the paper is accepted for publication.

---

### Decision · Program_Chairs · 2023-01-20

**Decision:**

Accept: poster

**Justification For Why Not Higher Score:**

We still have concerns (though minor) about the generalization of the proposed method to other networks and to larger datasets. Therefore, we cannot recommend it for a higher score.

**Justification For Why Not Lower Score:**

This work develops a strong and solid method for improving robustness using data augmentation alone, which is an important contribution and would be of interest to the general ICLR audience. Therefore, we should accept it.

**Metareview: Summary, Strengths And Weaknesses:**

This paper studies how to use data augmentation alone to improve adversarial training. The key finding is that data augmentation strategies should have strong diversity and well-balanced hardness to be effective. Two instantiations of such strategies, Cropshift and IDBH, are presented. Experiments on CIFAR-10, SVHN, and Tiny ImageNet are provided to demonstrate its effectiveness.

Overall, the reviewers found the proposed method to be well-motivated and the empirical results to be strong. However, they also had some concerns:  (1) confusion regarding the WideResNet results; (2) the absence of some strong baselines in Table 2; (3) the lack of ablations on the combination with regularization; (4) the overhead of heuristic search is huge. The rebuttal well addressed the concerns (1)-(3). Concern (4) was discussed in the AC-reviewer meeting and concluded to be minor. As a result, we recommend accepting this paper.

In the final version, the authors are strongly encouraged to discuss the generalization of their approach to other architectures, especially vision transformers, and to add ImageNet results if possible. They should also discuss potential plans to make heuristic search more efficient and effective. These additions will help future readers better understand the paper and improve its overall quality.


**Note From Pc:**

if the above contains the word "oral" or "spotlight" please see: "oral" presentation means -> notable-top-5% and "spotlight" means -> notable-top-25%. As stated in our emails, we are disassociating presentation type from AC recommendations

**Summary Of Ac-Reviewer Meeting:**

In the meeting, we first agreed that this work develops a strong and solid method for adversarial training in the typical setting,
e.g., small datasets like CIFAR + CNN architectures like ResNet18 or WideResNet. This is an important contribution that would be of interest to the general ICLR audience. Meanwhile, we also believe that the generalization of the method to other models, such as the recently emerged ViT architecture, and to larger datasets like ImageNet is a legitimate concern. However, we acknowledge that addressing these issues may be too challenging for this paper and could be considered minor. We strongly encourage the authors to discuss these issues in the final version of the paper.